Manuscript prepared for J. Name
with version 2.2 of the LaTeX class copernicus_discussions.cls.
Date: 4 August 2020

# Estimating the maximum of 363-day-smoothing highest 3-hourly $aa$ index in 3-day-interval by the preceding minimum of highest/lowest $aa$ value for the 11-year solar cycle

**Zhanle Du**

Key Laboratory of Solar Activity, National Astronomical Observatories, Chinese Academy of Sciences, Beijing 100012, China.

Correspondence to: Z. L. Du
(zldu@nao.cas.cn)

## Abstract

Predicting the strength of geomagnetic activity for an upcoming cycle is important in space weather service for planning future space missions. This study analyzed the **highest/lowest 3-hourly $aa$ index ($aa_H/aa_L$) in 3-day-interval, smoothed by 363 days to mimic the 13-month smoothing. It is found that the maximum of $aa_H$ ($aa_{Hmax}$) is well correlated to the preceding minimum of either $aa_H$ ($aa_{Hmin}, r = 0.85$) or $aa_L$ ($aa_{Lmin}, r = 0.89$) for the 11-year solar cycle. Based on these relationships, the strength of $aa_{Hmax}$ for cycle 25 is estimated to be $aa_{Hmax}(25) = 84.5 \pm 6.9$ (nT), about 30% stronger than that of cycle 24. This value is equivalent to the $Ap$ index of $Ap_{Hmax}(25) = 47.8 \pm 3.9 \pm 2.1$ (nT) if employing the high correlation between $Ap$ and $aa$ ($r = 0.93$), here $\pm 3.9$ and $\pm 2.1$ are derived from the uncertainty of $aa_{Hmax}(25)$ and the standard deviation of the fitting of $Ap$ to $aa$, respectively. The maximum of $aa_L$ ($aa_{Lmax}$) is also well correlated to the preceding $aa_{Hmin}(r = 0.80)$. The maximum of sunspot cycle ($R_m$) is much better correlated to the high geomagnetic activity ($aa_{Hmax}, r = 0.79$) than to the low one ($aa_{Lmax}, r = 0.37$). The rise time from $aa_{Hmin}$ to $aa_{Hmax}$** is weakly anti-correlated to the following $aa_{Hmax}$, $r = -0.42$. **Similar correlations are also found both for the 363-day-smoothing highest/lowest 3-hourly $Ap$ index in 3-day-interval and for the 13-month smoothed monthly mean $aa/Ap$ index. These results are expected to be useful in understanding the geomagnetic activity strength of solar cycle 25.**

## 1  Introduction

Studying and predicting geomagnetic activities are important in both geophysics and space weather. Severe geomagnetic activities may cause intense geomagnetic storms (Gonzalez et al., 1989, 1994; Chen et al., 2019), leading to disruptions in communication and deviations of **orbital motions of satellites(Yoshida and Yamagishi, 2010; Petrovay, 2020)**. With the current solar cycle 24 approaching its end, satellite and spacecraft-related departments want to know the strengths of both solar and geomag-

netic activities in the ensuing cycle 25 for planning future space missions.

Among various indices to quantitatively describe geomagnetic activities, the $aa$ index (Mayaud, 1972), derived from the 3-hourly K indices at two near-antipodal midlatitude stations in England and Australia, is the longest time series (since 1868) and has been widely used for analyzing long-term trends in the global geomagnetic activity (Russell and Mulligan, 1995; Marat et al., 2017; Du, 2011b; El-Borie et al., 2019) and for analyzing its correlation with both climate change (Cliver et al., 1998; Dobrica et al., 2009; Gavrilyeva et al., 2017) and solar activity (Echer et al., 2004; Prestes et al., 2006; Lukianova et al., 2009; Du, 2011a,c; Du and Wang, 2012; Singh and et al., 2019). The minimum $aa$ index ($aa_\mathrm{min}$), at or near the minimum of the solar cycle, has been widely used in predicting the maximum amplitude of the sunspot cycle ($R_\mathrm{m}$), the so-called Ohl's precursor method (Ohl, 1979; Brown and Williams, 1969; Du et al., 2009). But it is seldom used to directly predict the maximum $aa$ index ($aa_\mathrm{max}$) of an ensuing cycle.

The planetary geomagnetic index $Ap$ (Bartels, 1963) available since 1932, derived from the average of the measurements at 13 observatories around the globe, is a daily measure of the response of geomagnetic field to variations in the interplanetary magnetic field (IMF) and the solar wind (Li, 1997; McPherron, 1999; Tsurutani et al., 2006). It is the main global magnetic index forecasted by government agencies (McPherron, 1999). Most works on forecasting geomagnetic activity have been over short intervals, on the order of hours or days (McPherron, 1999; Abunina et al., 2013). In the earlier years, Kane (1988) pointed out that it is impossible to forecast the long-term geomagnetic activity through analyzing the daily, monthly and annual values of $Ap$ and $aa$ indices. Gordon (2015) demonstrated that long-term geomagnetic activity can only be predicted to within a limited threshold of accuracy due to the irregular trends and cycles in the annual data and nonlinear variability in the monthly series, through analyzing the $aa$ index.

**In this study, we analyze the relationship between the maximum $aa/Ap$ index and its preceding minimum for the 11-year solar cycle. First, in Sect. 2, we use the highest/lowest 3-hourly $aa$ index ($aa_\mathrm{H}/aa_\mathrm{L}$) in each 3 days' interval, smoothed by 363 days (121 points) to**

mimic the 13-month smoothing (Sect. 2.1). It is found that the maximum of $aa_H$ ($aa_{Hmax}$) is well correlated to the preceding minimum of either $aa_H$ ($aa_{Hmin}$) or $aa_L$ ($aa_{Lmin}$) for the 11-year solar cycle (Sect. 2.2), which can be used to estimate $aa_{Hmax}$ of the ensuing cycle. The maximum of $aa_L$ ($aa_{Lmax}$) is also found to be well correlated to the preceding $aa_{Hmin}$ (Sect. 2.3). The rise time of $aa_H$ from $aa_{Hmin}$ to $aa_{Hmax}$ is only weakly anti-correlated to the following $aa_{Hmax}$ (Sect. 2.4). Similar results are analyzed by using the 363-day-smoothing highest and lowest 3-hourly $Ap$ indices in each 3 days' interval (Sect. 3) and by using the 13-month smoothed (with half weight at the two ends) monthly mean $aa$ and $Ap$ indices (Sect. 4). Some conclusions are discussed and summarized in Sect. 5.

## 2    Result for 363-day-smoothing highest/lowest 3-hourly $aa$ index in 3-day-interval

### 2.1    Data

First in this section, we use the **3-hourly** $aa$ index since 1868 (to 2020 **August 1st**) from the International Service of Geomagnetic Indices (ISGI)[1]. **We find out the highest/lowest** $aa$ index ($aa_H/aa_L$) **from 24 values of the 3-hourly** $aa$ **indices in each 3 days' interval. In order to reduce accidental events in the data, both** $aa_H$ **and** $aa_L$ **are smoothed by 363 days (121 points) to mimic the 13-month smoothing, as shown in Fig. 1 (solid). The 13-month smoothed monthly mean International sunspot number series (**$R_I$, **Clette et al., 2016) of the second version[2] is used for comparison (dotted).**

In the upper panel of the figure, the dashed (dash-dotted) line indicates the maximum (minimum) of $aa_H$, $aa_{Hmax}$ ($aa_{Hmin}$). While in the lower panel, the dashed (dash-dotted) line indicates the maximum (minimum) of $aa_L$, $aa_{Lmax}$ ($aa_{Lmin}$). These parameters are displayed in Table 1, in which $T_{Hr}$ is the rise time of $aa_H$ from $aa_{Hmin}$ to $aa_{Hmax}$, and $R_m$ the maximum of $R_I$ for the 11-year solar cycle. The last row denotes the averages of the parameters.

---

[1]http://isgi.unistra.fr/
[2]http://www.sidc.be/silso/datafiles

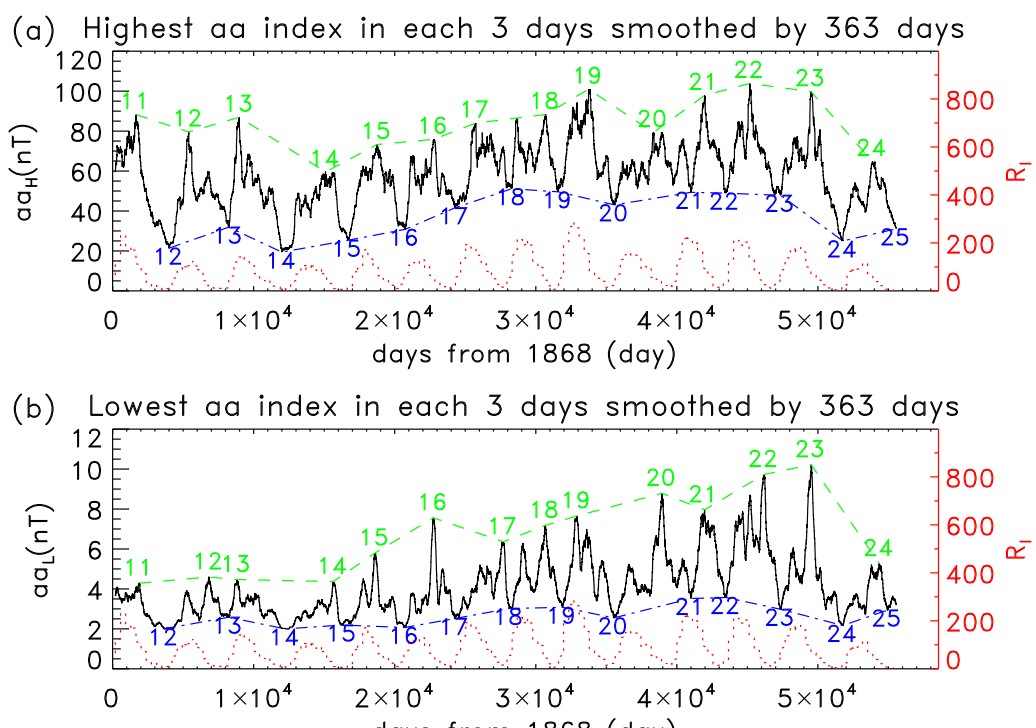

**Fig. 1.** (a) The highest ($aa_{\mathrm{H}}$) and (b) the lowest ($aa_{\mathrm{L}}$) 3-hourly $aa$ index in each 3 days (black solid), smoothed by 121 points (363 days). The numbers in the figure indicate the 11-year solar cycles. The upper dashed and lower dash-dotted lines indicate the maxima and minima, respectively, for the 11-year solar cycle. The red dotted line represents the 13-month smoothed monthly mean sunspot number ($R_{\mathrm{I}}$) for comparison.

**Table 1.** The minimum ($aa_{Hmin}$) and maximum ($aa_{Hmax}$) of 363-day-smoothing highest 3-hourly $aa$ index ($aa_H$) in each 3 days, the rise time of $aa_{Hmax}$ ($T_{Hr}$), the minimum ($aa_{Lmin}$) and maximum ($aa_{Lmax}$) of 363-day-smoothing lowest 3-hourly $aa$ index ($aa_L$) in each 3 days, and the maximum ($R_m$) of 13-month smoothed monthly mean sunspot number for solar cycles 11-25.

| $n$ | $aa_{Hmin}$(nT) | $aa_{Hmax}$(nT) | $T_{Hr}$(yr) | $aa_{Lmin}$(nT) | $aa_{Lmax}$(nT) | $R_m$ |
|---|---|---|---|---|---|---|
| 11 | | 88.24 | | | 4.29 | 234.0 |
| 12 | 21.41 | 79.60 | 3.81 | 2.00 | 4.60 | 124.4 |
| 13 | 31.88 | 86.92 | 2.08 | 2.58 | 4.47 | 146.5 |
| 14 | 19.40 | 59.55 | 8.47 | 2.00 | 4.37 | 107.1 |
| 15 | 25.31 | 73.28 | 5.72 | 2.19 | 5.80 | 175.7 |
| 16 | 30.80 | 75.90 | 5.56 | 2.07 | 7.56 | 130.2 |
| 17 | 41.60 | 84.01 | 4.01 | 2.52 | 6.36 | 198.6 |
| 18 | 51.13 | 88.31 | 6.57 | 3.02 | 7.18 | 218.7 |
| 19 | 49.60 | 100.99 | 5.89 | 3.08 | 7.66 | 285.0 |
| 20 | 43.08 | 79.31 | 7.54 | 2.55 | 8.79 | 156.6 |
| 21 | 49.35 | 97.87 | 2.63 | 3.52 | 7.96 | 232.9 |
| 22 | 48.93 | 103.89 | 4.78 | 3.57 | 9.73 | 212.5 |
| 23 | 47.84 | 99.75 | 5.92 | 2.98 | 10.22 | 180.3 |
| 24 | 25.02 | 64.81 | 5.73 | 2.17 | 5.27 | 116.4 |
| 25 | 31.05 | | | 2.94 | | |
| Av. | 36.89 | 84.46 | 5.29 | 2.66 | 6.73 | 179.9 |

The correlation coefficients between the parameters in Table 1 are listed in Table 2 for comparison. It is seen in Table 2 that $R_m$ is well correlated to $aa_{Hmin}(r=0.84)$, $aa_{Hmax}(r=0.79)$, $aa_{Lmin}(r=0.81)$, and positively correlated to $aa_{Lmax}(r=0.37)$. It implies that the stronger the solar activity ($R_l$), the higher the geomagnetic activity ($aa$). But the maximum of sunspot cycle ($R_m$) is much better correlated to the high geomagnetic activity ($aa_{Hmax}, r=0.79$) than to the low one ($aa_{Lmax}, r=0.37$), implying that the low geo-

**Table 2.** Correlation coefficients between parameters in Table 1.

| $x/y$ | $aa_{Hmin}$ | $aa_{Hmax}$ | $T_{Hr}$ | $aa_{Lmin}$ | $aa_{Lmax}$ | $R_m$ |
|---|---|---|---|---|---|---|
| $aa_{Hmin}$ | 1.00 | 0.85 | $-0.10$ | 0.85 | 0.80 | 0.84 |
| $aa_{Hmax}$ | 0.85 | 1.00 | $-0.42$ | 0.89 | 0.63 | 0.79 |
| $T_{Hr}$ | $-0.10$ | $-0.42$ | 1.00 | $-0.28$ | 0.13 | $-0.18$ |
| $aa_{Lmin}$ | 0.85 | 0.89 | $-0.28$ | 1.00 | 0.70 | 0.81 |
| $aa_{Lmax}$ | 0.80 | 0.63 | 0.13 | 0.70 | 1.00 | 0.37 |

magnetic activity depends less on the solar activity than the high one does. The correlation between $R_m$ and $aa_{Hmin}$ (or $aa_{Lmin}$) is related to the Ohl's precursor method (Ohl, 1979) for predicting $R_m$. Some other correlations are analyzed to estimate $aa_{Hmax}$ (Sect. 2.2), $aa_{Lmax}$ (Sect. 2.3), and $T_{Hr}$ (Sect. 2.4) in this section.

5 ## 2.2 Relationship between $aa_{Hmax}$ and its preceding $aa_{Hmin}/aa_{Lmin}$

One can see in Table 2 that $aa_{Hmax}$ is well correlated to its preceding $aa_{Hmin}(r = 0.85)$ and $aa_{Lmin}(r = 0.89)$, as shown in Fig. 2 for the scatter plots of $aa_{Hmax}$ against $aa_{Hmin}$(a) and $aa_{Lmin}$(b). The solid line represents the linear fit of $aa_{Hmax}$ to $aa_{Hmin}$ ($aa_{Lmin}$) with the least-squares-fit regression equation given by

$$\begin{cases} aa_{Hmax} = 47.1 \pm 7.1 + (0.99 \pm 0.18)aa_{Hmin}, \ \sigma = 7.3, \\ aa_{Hmax} = 25.3 \pm 9.4 + (22.3 \pm \ \ 3.5)aa_{Lmin}, \ \sigma = 6.5, \end{cases} \tag{1}$$

10 where $\pm$ indicates the 1-$\sigma$ deviation of the fitting coefficient and $\sigma$ the standard deviation of the fitting.

Based on the above relationships, the $aa$ index at the minimum can be used as an indicator to estimate the following maximum. One can estimate $aa_{Hmax}$ for cycle $n = 25$ by

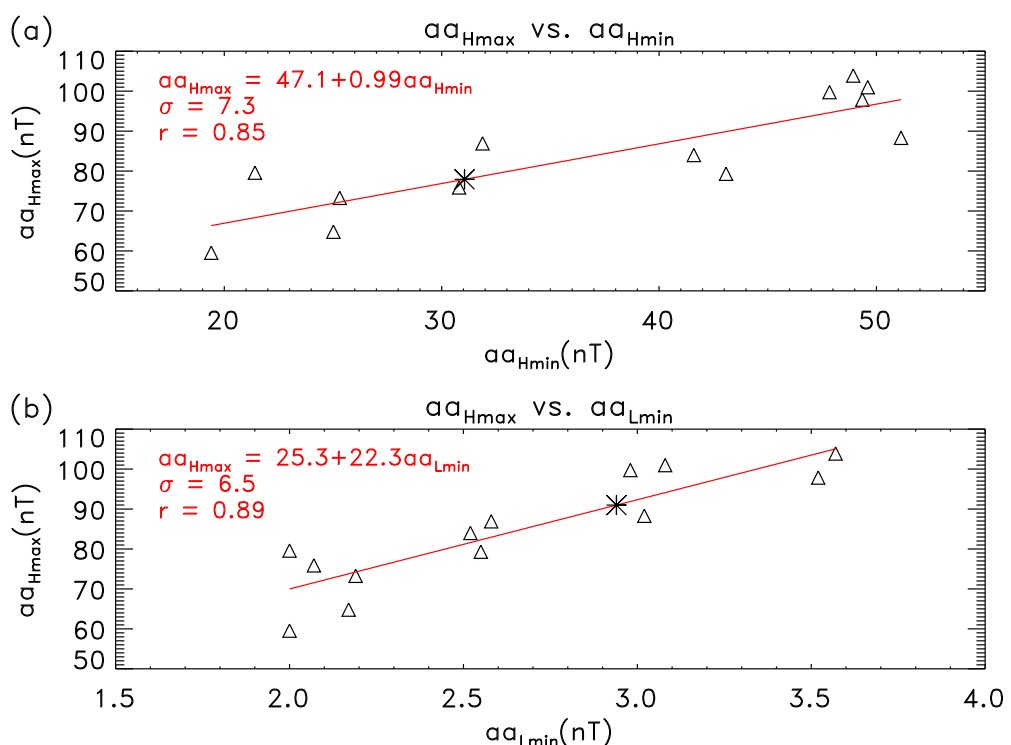

**Fig. 2.** (a) Scatter plot of $aa_{Hmax}$ against $aa_{Hmin}$ (triangles) and the linear fit (solid). (b) Scatter plot of $aa_{Hmax}$ against $aa_{Lmin}$ (triangles) and the linear fit (solid).

**substituting the values of $aa_{Hmin}$(31.05 nT) and $aa_{Lmin}$(2.94 nT) into the above equations,**

$$\begin{cases} aa_{Hmax1}(25) = 77.9 \pm 7.3 (nT), \text{ from } aa_{Hmin}, \\ aa_{Hmax2}(25) = 91.0 \pm 6.5 (nT), \text{ from } aa_{Lmin}, \end{cases} \tag{2}$$

(labelled by asterisk). As these values are derived by the fitting equations (Eq.(1)) with similar correlation coefficients (0.85 and 0.89), we take their average,

$$aa_{\mathsf{Hmax}}(25) = \frac{1}{2}[aa_{\mathsf{Hmax1}}(25) + aa_{\mathsf{Hmax2}}(25)]$$
$$= 84.5 \pm 6.9 (\mathsf{nT}), \tag{3}$$

as an estimate of $aa_{\mathsf{Hmax}}(25)$. **It implies that the 363-day-smoothing highest 3-hourly** $aa$
**index in 3-day-interval during the maximum period of cycle 25 is estimated to be close to the average (84.46 nT) over the past cycles (Table 1), but higher than that (64.81 nT) of cycle 24 by about 30.3%.**

It should be pointed out that the above estimate may be an upper limit of $aa_{\mathsf{Hmax}}(25)$
as the values of $aa_{\mathsf{Hmin}}(25)$ and $aa_{\mathsf{Lmin}}(25)$ may not be finally determined (usually about
10 one year after the minimum of a solar cycle (24)). Although we are not quite sure if the current $aa_{\mathsf{H}}$ ($aa_{\mathsf{L}}$), 31.05 (3.19) in January 2020, would decrease to a smaller value than that, 31.05 (2.94), used in the current work, there would not be significant variations in $aa_{\mathsf{Hmin}}$, $aa_{\mathsf{Lmiin}}$ and the above estimate, because the sunspot number ($R_{\mathsf{I}}$) shows a sign to stop decreasing and to oscillate around the minimum during the recent few months.

## 2.3   Relationship between $aa_{\mathsf{Lmax}}$ and the preceding $aa_{\mathsf{Hmin}}$

**One can also see in Table 2 that** $aa_{\mathsf{Lmax}}$ **is well correlated to the preceding** $aa_{\mathsf{Hmin}}(r = 0.80)$
**or** $aa_{\mathsf{Lmin}}(r = 0.70)$. **Fig. 3(a) shows the scatter plot of** $aa_{\mathsf{Lmax}}$ **against** $aa_{\mathsf{Hmin}}$. **The linear fitting equation of** $aa_{\mathsf{Lmax}}$ **to** $aa_{\mathsf{Hmin}}$ **(solid) is**

$$aa_{\mathsf{Lmax}} = 2.0 \pm 1.2 + (0.131 \pm 0.030)aa_{\mathsf{Hmin}}, \ \sigma = 1.2. \tag{4}$$

**Substituting** $aa_{\mathsf{Hmin}}(25) = 31.05$ **(nT) into this equation, one can estimate the 363-day-smoothing** *lowest* **3-hourly** $aa$ **index in 3-day-interval during the maximum period of cycle 25,** $aa_{\mathsf{Lmax}} = 6.1 \pm 1.2$ **(nT). This value is slightly lower than the average (6.73 nT) over the past cycles, but higher than that (5.27 nT) of cycle 24 by 15.7%.**

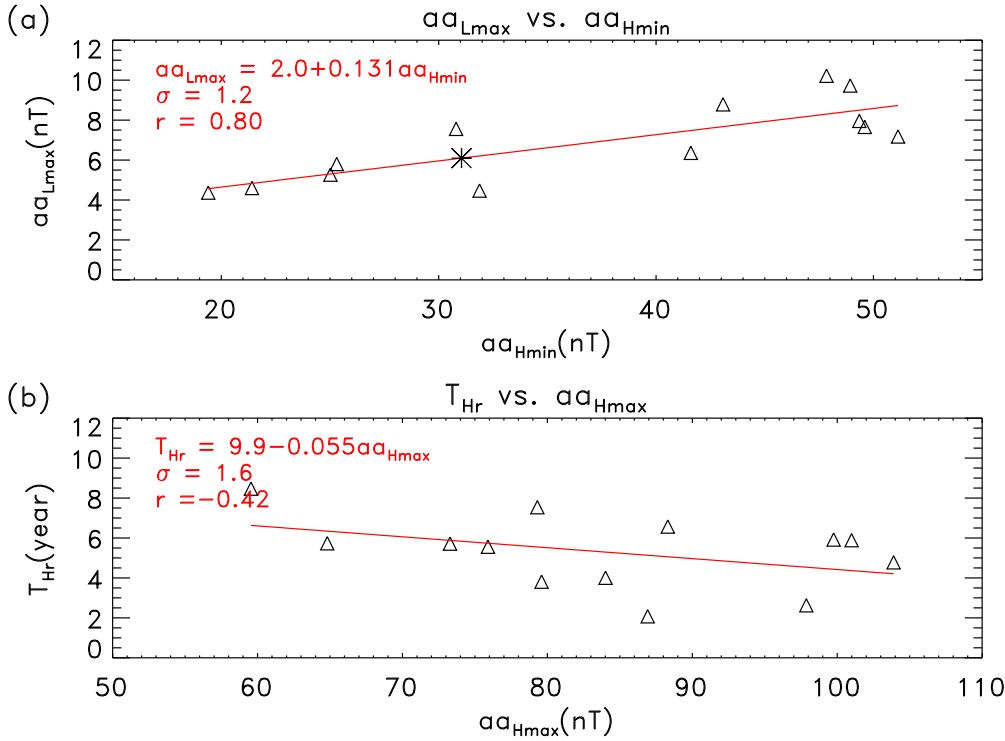

**Fig. 3.** (a) Scatter plot of $aa_{Lmax}$ against $aa_{Hmin}$ (triangles) and the linear fit (solid). (b) Scatter plot of $T_{Hr}$ against $aa_{Hmax}$ (triangles) and the linear fit (solid).

## 2.4 Relationship between the rise time and following maximum

Now, we analyze if the rise time of the $aa$ geomagnetic index for the 11-year solar cycle **is correlated to** the following maximum **so that it can be used to estimate the rise time**, as

the case often used in the solar (sunspot) cycle (Waldmeier, 1939).

**Figure 3(b) shows the scatter plot of the rise time ($T_{Hr}$) from $aa_{Hmin}$ to $aa_{Hmax}$ for the 11-year solar cycle against its following maximum ($aa_{Hmax}$). The solid line indicates the linear fit of $T_{Hr}$ to $aa_{Hmax}$ by the following fitting equation,**

$$T_{Hr} = 9.9 \pm 3.0 - (0.055 \pm 0.036)aa_{Hmax}, \ \sigma = 1.6 (\text{years}). \tag{5}$$

**The anti-correlation coefficient between $T_{Hr}$ and $aa_{Hmax}$, $r = -0.42$ (at a confidence level of about 84%), is so weak that it can hardly be used to estimate the rise time ($T_{Hr}$). If the rise time is computed from the minimum of sunspot activity ($R_l$) to $aa_{Hmax}$, the correlation is even weaker, $r = -0.14$.**

## 3 Result for 363-day-smoothing highest/lowest 3-hourly $Ap$ index in 3-day-interval

Then in this section, we analyze the previous result using the 3-hourly $Ap$ index since 1932[3] (available to 2018 April). Similar to the $aa$ index, we find out the highest/lowest 3-hourly $Ap$ index ($Ap_H/Ap_L$) in each 3 days' interval. Both $Ap_H$ and $Ap_L$ are smoothed by 363 days (121 points), as shown in Fig. 4 (solid). Table 3 displays the maximum/minimum of $Ap_H$ ($Ap_{Hmax}/Ap_{Hmin}$), the rise time of $Ap_H$ from $Ap_{Hmin}$ to $Ap_{Hmax}$ ($T_{Ha}$), and the maximum/minimum of $Ap_L$ ($Ap_{Lmax}/Ap_{Lmin}$).

The correlation coefficients between the parameters in Table 3 are listed in Table 4. One can see in Table 4 that $R_m$ is well correlated to $Ap_{Hmin}(r = 0.88)$, $Ap_{Hmax}(r = 0.73)$, $Ap_{Lmin}(r = 0.82)$, and positively correlated to $Ap_{Lmax}(r = 0.35)$, similar to the case for the $aa$ index. Some other significant correlations are analyzed in this section.

### 3.1 Relationship between $Ap_{Hmax}$ and the preceding $Ap_{Hmin}/Ap_{Lmin}$

It is seen in Table 4 that $Ap_{Hmax}$ is well correlated to the preceding $Ap_{Hmin}(r = 0.96)$ and $Ap_{Lmin}(r = 0.79)$, as shown in Fig. 5 for the scatter plots of $Ap_{Hmax}$ against $Ap_{Hmin}$(a)

---

[3]http://www.gfz-potsdam.de/en/kp-index

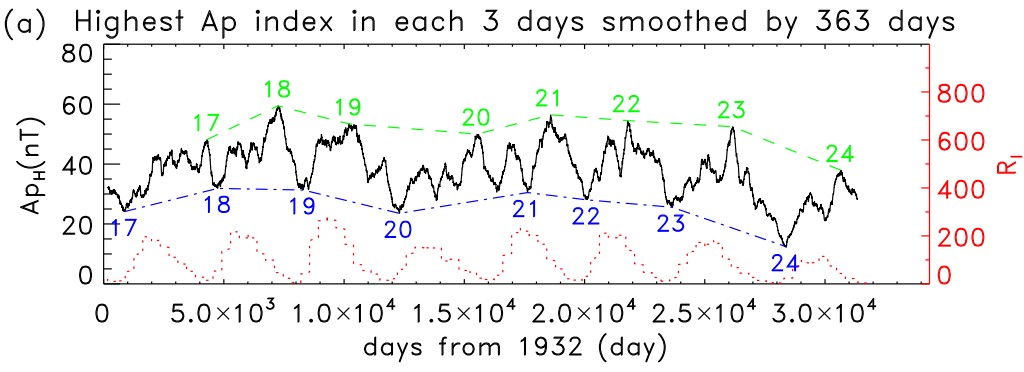

(a) Highest Ap index in each 3 days smoothed by 363 days

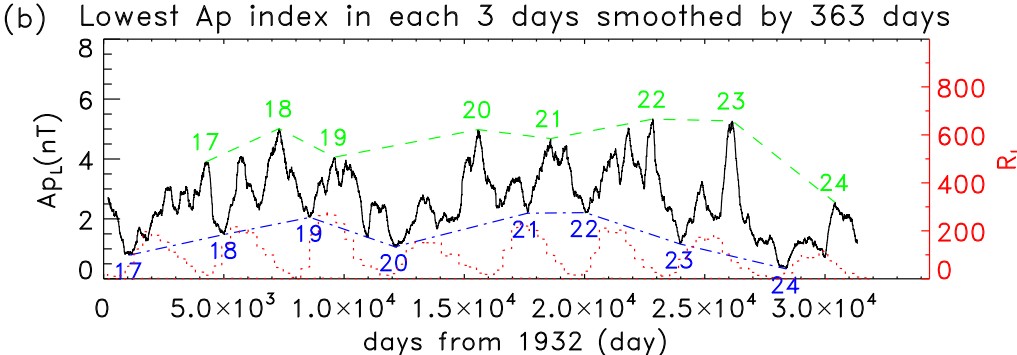

(b) Lowest Ap index in each 3 days smoothed by 363 days

**Fig. 4.** Similar to Fig. 1 for the $Ap$ index: (a) the highest $Ap$ index ($Ap_\mathrm{H}$) and (b) the lowest $Ap$ index ($Ap_\mathrm{L}$) in each 3 days (black solid), smoothed by 121 points (363 days).

and $Ap_\mathrm{Lmin}$(b). The linear fitting equation between $Ap_\mathrm{Hmax}$ and $Ap_\mathrm{Hmin}$ ($Ap_\mathrm{Lmin}$) is

$$\begin{cases} Ap_\mathrm{Hmax} = 25.9 \pm 3.0 + (0.99 \pm 0.11)Ap_\mathrm{Hmin}, \ \sigma = 1.8, \\ Ap_\mathrm{Hmax} = 41.1 \pm 3.7 + \ \ (7.4 \pm \ \ 2.4)Ap_\mathrm{Lmin}, \ \sigma = 4.0. \end{cases} \tag{6}$$

**Table 3.** The minimum ($Ap_{Hmin}$) and maximum ($Ap_{Hmax}$) of 363-day-smoothing highest 3-hourly $Ap$ index ($Ap_H$) in each 3 days, the rise time of $Ap_{Hmax}$ ($T_{Ha}$), the minimum ($Ap_{Lmin}$) and maximum ($Ap_{Lmax}$) of 363-day-smoothing lowest 3-hourly $Ap$ index ($Ap_L$) in each 3 days, and the maximum ($R_m$) of 13-month smoothed sunspot number for solar cycles 17-24.

| $n$ | $Ap_{Hmin}$(nT) | $Ap_{Hmax}$(nT) | $T_{Ha}$(yr) | $Ap_{Lmin}$(nT) | $Ap_{Lmax}$(nT) | $R_m$ |
|-----|-----------------|-----------------|--------------|-----------------|-----------------|-------|
| 17  | 24.15 | 48.19 | 9.40 | 0.79 | 3.91 | 198.6 |
| 18  | 31.86 | 59.43 | 6.86 | 1.48 | 5.01 | 218.7 |
| 19  | 31.34 | 53.40 | 5.22 | 2.05 | 4.06 | 285.0 |
| 20  | 23.55 | 50.01 | 8.89 | 1.07 | 4.98 | 156.6 |
| 21  | 30.47 | 56.45 | 2.62 | 2.19 | 4.68 | 232.9 |
| 22  | 28.03 | 54.50 | 4.60 | 2.21 | 5.33 | 212.5 |
| 23  | 25.58 | 52.48 | 6.89 | 1.16 | 5.27 | 180.3 |
| 24  | 12.39 | 37.92 | 6.23 | 0.33 | 2.55 | 116.4 |
| Av. | 25.92 | 51.55 | 6.34 | 1.41 | 4.47 | 200.1 |

**Table 4.** Correlation coefficients between parameters in Table 3.

| $x/y$ | $Ap_{Hmin}$ | $Ap_{Hmax}$ | $T_{Ha}$ | $Ap_{Lmin}$ | $Ap_{Lmax}$ | $R_m$ |
|-------|-------------|-------------|----------|-------------|-------------|-------|
| $Ap_{Hmin}$ | 1.00  | 0.96  | −0.34 | 0.83  | 0.70  | 0.88  |
| $Ap_{Hmax}$ | 0.96  | 1.00  | −0.33 | 0.79  | 0.82  | 0.73  |
| $T_{Ha}$    | −0.34 | −0.33 | 1.00  | −0.72 | −0.09 | −0.44 |
| $Ap_{Lmin}$ | 0.83  | 0.79  | −0.72 | 1.00  | 0.59  | 0.82  |
| $Ap_{Lmax}$ | 0.70  | 0.82  | −0.09 | 0.59  | 1.00  | 0.35  |

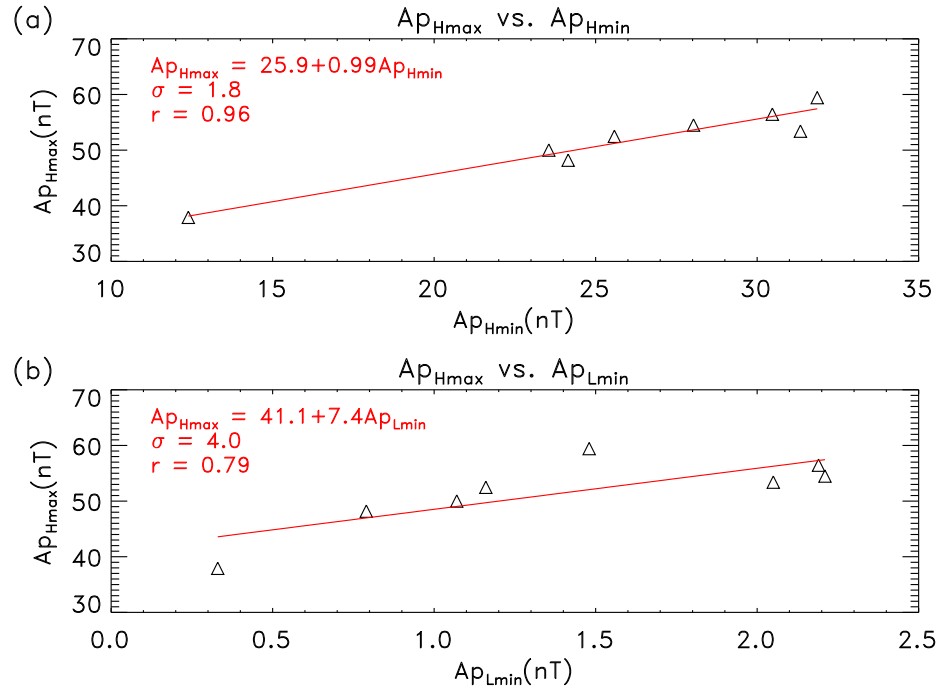

**Fig. 5.** (a) Scatter plot of $Ap_{\mathrm{Hmax}}$ against $Ap_{\mathrm{Hmin}}$ (triangles) and the linear fit (solid). (b) Scatter plot of $Ap_{\mathrm{Hmax}}$ against $Ap_{\mathrm{Lmin}}$ (triangles) and the linear fit (solid).

If the values of $Ap_{\mathrm{Hmin}}$ and $Ap_{\mathrm{Lmin}}$ are obtained in advance, the value of $Ap_{\mathrm{Hmax}}$ can be estimated from the above equations. However, these values are unknown at present as the $Ap$ index is available only to 2018 April.

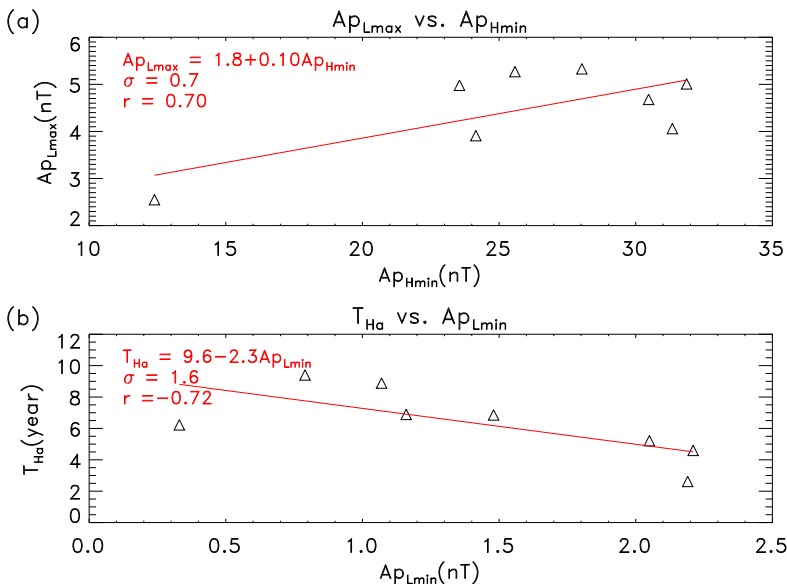

**Fig. 6.** (a) Scatter plot of $Ap_{\text{Lmax}}$ against $Ap_{\text{Hmin}}$ (triangles) and the linear fit (solid). (b) Scatter plot of $T_{\text{Ha}}$ against $Ap_{\text{Lmin}}$ (triangles) and the linear fit (solid).

## 3.2  Relationship between $Ap_{\text{Lmax}}$ and the preceding $Ap_{\text{Hmin}}$

It is also seen in Table 4 that $Ap_{\text{Lmax}}$ is well correlated to the preceding $Ap_{\text{Hmin}}(r = 0.70)$, as shown in Fig. 6(a). The linear fitting equation of $Ap_{\text{Lmax}}$ to $Ap_{\text{Hmin}}$ (solid) is

$$Ap_{\text{Lmax}} = 1.8 \pm 1.1 + (0.10 \pm 0.04)Ap_{\text{Hmin}}, \ \sigma = 0.7. \tag{7}$$

5  If $Ap_{\text{Hmin}}$ is known, $Ap_{\text{Lmax}}$ can be estimated from this equation.

### 3.3  Relationship between the rise time and preceding minimum

One may note in Table 4 that the anti-correlation between the rise time ($T_{\text{Ha}}$) from $Ap_{\text{Hmin}}$ to $Ap_{\text{Hmax}}$ and the following maximum ($Ap_{\text{Hmax}}$) is very weak, $r = -0.33$. While the anti-correlation between $T_{\text{Ha}}$ and the preceding minimum ($Ap_{\text{Lmin}}$) is strong, $r = -0.72$ (at the 95% confidence level). Figure 6(b) shows the scatter plot of $T_{\text{Ha}}$ against $Ap_{\text{Lmin}}$, fitted by the following linear equation,

$$T_{\text{Ha}} = 9.6 \pm 1.4 - (2.3 \pm 0.9)Ap_{\text{Lmin}}, \; \sigma = 1.6(\text{years}). \tag{8}$$

Similarly, if $Ap_{\text{Lmin}}(25)$ is known, $T_{\text{Ha}}$ for cycle 25 can be estimated from this equation.

### 3.4  Relationship between $Ap$ and $aa$

Now, we analyze the relationship between the $Ap$ and $aa$ indices, as shown in Fig. 7(a) for the scatter plot of the 363-day-smoothing 3-hourly $Ap$ against $aa$ indices since 1932 (dots). The solid line represents the linear fit of $Ap$ to $aa$ with the least-squares-fit regression equation given by

$$Ap = 0.12 \pm 0.01 + (0.5647 \pm 0.0005)aa, \; \sigma = 2.1. \tag{9}$$

The correlation coefficient between the fitted and observed values is $r = 0.93$ (or 0.75 if using the non-smoothed series) at a confidence level greater than 99%. It is obvious that $Ap$ is highly correlated with $aa$, as they are based on the same observations.

According to this equation, the maximum of 363-day smoothing highest $Ap$ value for cycle 25 can be estimated by substituting the estimated $aa_{\text{Hmax}}(25) = 84.5 \pm 6.9$ (nT) in Sect. 2.2 into this equation, $Ap_{\text{Hmax}}(25) = 47.8 \pm 3.9 \pm 2.1$ (nT), here $\pm 3.9$ is derived from the uncertainty ($\pm 6.9$) of $aa_{\text{Hmax}}(25)$ and $\pm 2.1$ is the standard deviation of the fitting of $Ap$ to $aa$.

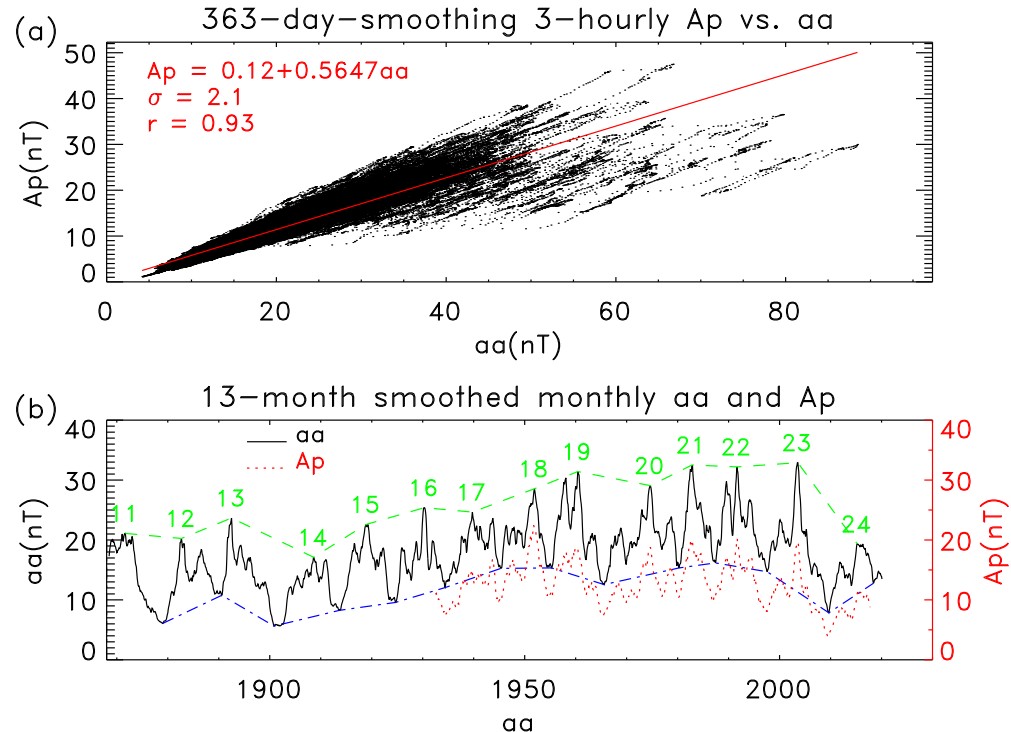

**Fig. 7.** (a) Scatter plot of the 363-day-smoothing 3-hourly $Ap$ against $aa$ indices since 1932 (dots) and the linear fit (solid). (b) The 13-month smoothed monthly mean time series of $aa$ (solid) since 1868 and $Ap$ (dotted) since 1932. The numbers in the figure indicate the solar cycles. The upper dashed and lower dash-dotted lines represent the maxima ($aa_{max}$) and minima ($aa_{min}$) of $aa$, respectively, for the 11-year solar cycle.

**Table 5.** Parameters of 13-month smoothed monthly mean $aa$ and $A_\mathrm{p}$ for the solar cycle.

| $n$ | $aa_{min}$(nT) | $aa_{max}$(nT) | $T_r$(month) | $Ap_{min}$(nT) | $Ap_{max}$(nT) | $T_a$(month) |
|---|---|---|---|---|---|---|
| 11 | | 21.10 | | | | |
| 12 | 6.07 | 20.25 | 44 | | | |
| 13 | 10.77 | 23.66 | 23 | | | |
| 14 | 5.64 | 17.12 | 93 | | | |
| 15 | 8.26 | 22.60 | 63 | | | |
| 16 | 9.57 | 25.39 | 68 | | | |
| 17 | 12.06 | 24.66 | 64 | 7.29 | 16.82 | 112 |
| 18 | 15.26 | 28.56 | 79 | 9.78 | 22.45 | 82 |
| 19 | 15.34 | 31.42 | 62 | 10.55 | 18.64 | 56 |
| 20 | 12.56 | 29.07 | 109 | 7.37 | 18.81 | 111 |
| 21 | 15.33 | 32.51 | 31 | 10.37 | 20.08 | 29 |
| 22 | 16.18 | 32.20 | 51 | 9.62 | 20.24 | 57 |
| 23 | 14.69 | 32.90 | 72 | 8.11 | 19.65 | 72 |
| 24 | 7.85 | 19.33 | 67 | 3.84 | 11.72 | 71 |
| 25 | 12.78 | | | | | |
| Av. | 11.60 | 25.77 | 63.5 | 8.41 | 18.55 | 73.8 |

## 4  Result for the 13-month smoothed monthly mean $aa/Ap$ index

**At last in this section, we simply analyze the previous result using the 13-month smoothed (with half weight at the two ends) monthly mean $aa$ index (solid) since 1868 and $Ap$ index (dotted) since 1932, as shown in Fig. 7(b).** The upper dashed and lower dash-dotted lines represent the maximum ($aa_{max}$) and minimum ($aa_{min}$) of the $aa$ index, respectively, for the 11-year solar cycle. The parameters are listed in Table 5, in which, $T_r$ is the rise time from $aa_{min}$ to $aa_{max}$, $Ap_{max}$ and $Ap_{min}$ are the maximum and minimum of the $Ap$

index for the 11-year solar cycle, respectively, and $T_a$ is the rise time from $Ap_{min}$ to $Ap_{max}$.

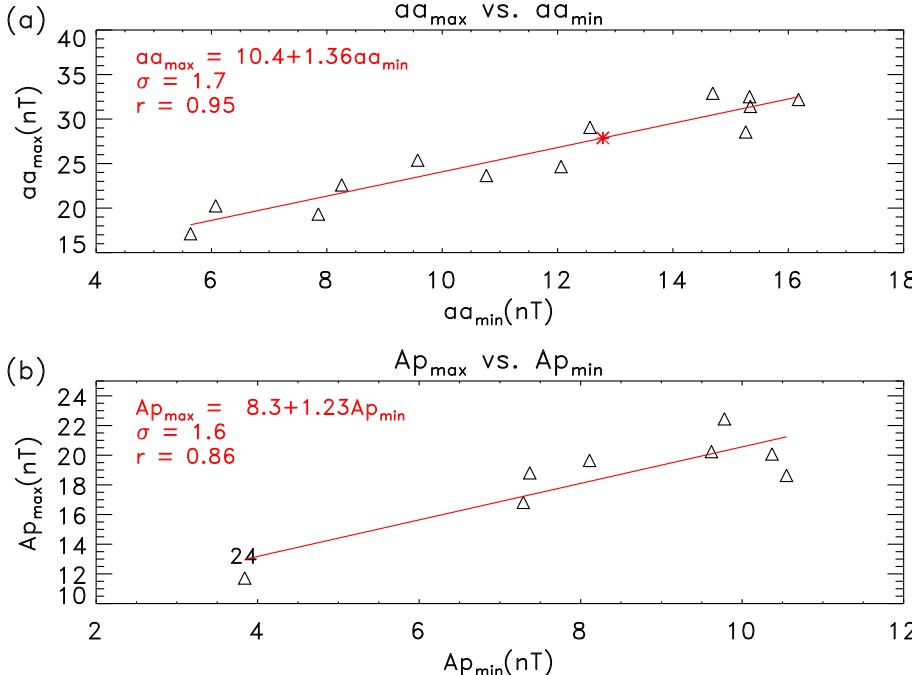

**Fig. 8.** (a) Scatter plot of $aa_{max}$ against $aa_{min}$ (triangles) and the linear fit (solid). (b) Scatter plot of $Ap_{max}$ against $Ap_{min}$ (triangles) and the linear fit (solid).

## 4.1 Relationship between $aa_{max}$ and $aa_{min}$

Figure. 8(a) shows the scatter plot of $aa_{max}$ against $aa_{min}$ for cycles 11-24 (triangles). The solid line indicates the linear fit of $aa_{max}$ to $aa_{min}$ by the following equation,

$$aa_{max} = 10.4 \pm 1.7 + (1.36 \pm 0.14)aa_{min}, \; \sigma = 1.7. \tag{10}$$

The correlation coefficient between $aa_{max}$ and $aa_{min}$ is $r = 0.95$ (at a confidence level greater than 99%), **slightly higher than that,** $r = 0.85(0.89)$**, for the correlation between** $aa_{Hmax}$ **and** $aa_{Hmin}(aa_{Lmin})$ **in Fig. 2 using the 363-day-smoothing highest (lowest) 3-hourly** $aa$ **index in 3-day-interval.**

Substituting the value of $aa_{min}$ (12.78) for cycle $n = 25$ into this equation, one can estimate $aa_{max}(25) = 27.9 \pm 1.7$ (asterisk), about 44% higher than that (19.33) of cycle 24. **This estimate is similar to the case in Sect. 2.2 that the estimate (91.0) of** $aa_{Hmax}(25)$ **from** $aa_{Lmin}$ **in Eq. (2) is about 40% higher than that (64.81 nT) of cycle 24 using the minimum of 363-day-smoothing lowest 3-hourly** $aa$ **index in 3-day-interval.**

## 4.2 Relationship between $Ap_{max}$ and $Ap_{min}$

Figure 8(b) illustrates the scatter plot of $Ap_{max}$ against $Ap_{min}$ for cycles 17-24 (triangles). It is seen in the figure that $Ap_{max}$ is also well correlated to $Ap_{min}$, with a correlation coefficient of $r = 0.86$ (at a confidence level greater than 99%), **slightly lower (higher) than that,** $r = 0.96(0.79)$**, for the correlation between** $Ap_{Hmax}$ **and** $Ap_{Hmin}(Ap_{Lmin})$ **in Fig. 5 using the 363-day-smoothing highest (lowest) 3-hourly** $Ap$ **index in 3-day-interval.** The linear fitting equation of $Ap_{max}$ to $Ap_{min}$ (solid) is

$$Ap_{max} = 8.3 \pm 2.5 + (1.23 \pm 0.29)Ap_{min}, \; \sigma = 1.6. \tag{11}$$

If $Ap_{min}$ is known, $Ap_{max}$ can be estimated from this equation.

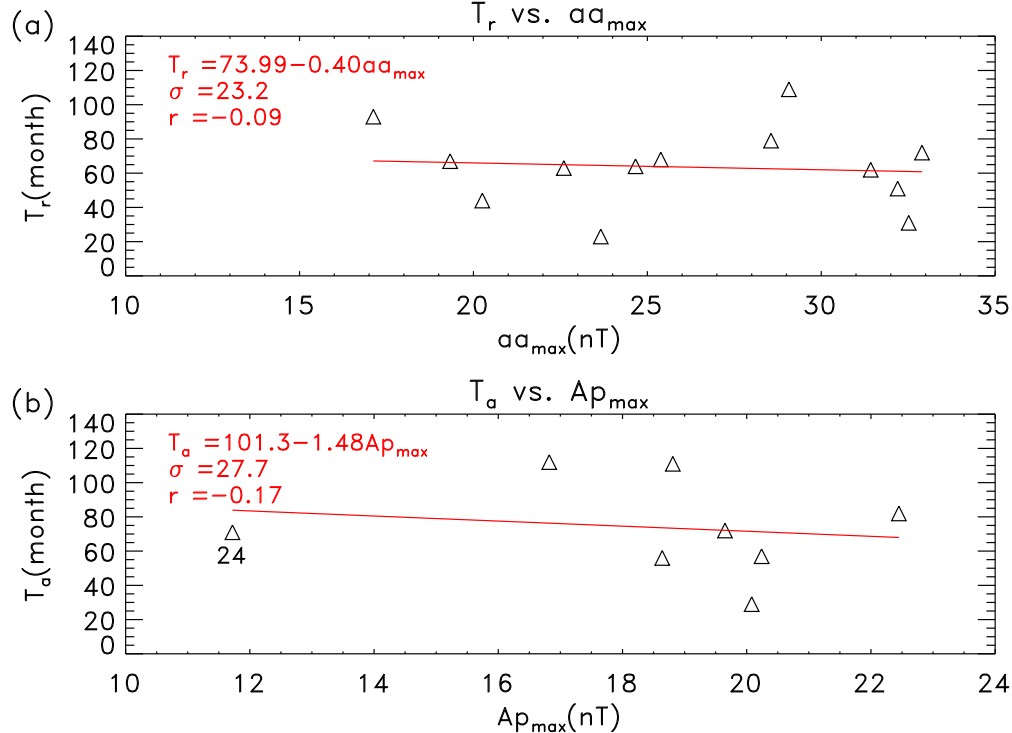

**Fig. 9.** (a) Scatter plot of $T_a$ against $aa_{max}$ (triangles) and the linear fit (solid). (b) Scatter plot of $T_r$ against $Ap_{max}$ (triangles) and the linear fit (solid).

## 4.3 Relationship between the rise time and following maximum

Figure 9(a) shows the scatter plot of the rise time ($T_r$) of $aa_{max}$ from $aa_{min}$ to $aa_{max}$ against the maximum ($aa_{max}$). It is seen in this figure that the data points are much

scattered, and so $T_r$ is nearly uncorrelated to the following $aa_{\max}$, $r = -0.09$. Similarly, the rise time ($T_a$) of $Ap_{\max}$ from $Ap_{\min}$ to $Ap_{\max}$ is also nearly uncorrelated to the following maximum ($Ap_{\max}$), $r = -0.17$, as shown in Fig. 9(b) for the scatter plot of $T_a$ against $Ap_{\max}$. Therefore, these correlations are unable to be used to estimate the rise times of $aa_{\max}$ and $Ap_{\max}$.

## 5 Discussions and Conclusions

**It is well known that the aa index is positively correlated to the solar activity (as represented by $R_l$), since the latter is the main source of the former (Legrand and Simon, 1981; Feynman, 1982; Echer et al., 2004). In general, the stronger the solar activity, the higher the ($aa$) geomagnetic activity. However the relationship between $aa$ and $R_l$ is not a simple linear one (Borello-Filisetti et al., 1992; Mussino et al., 1994; Kishcha et al., 1999; Lockwood et al., 1999; Echer et al., 2004; Tsurutani et al., 2006; Du, 2011a,c, 2020). The $aa$ index tends to lag behind $R_l$ about 2–3 years around a solar cycle maximum (Wang et al., 2000; Echer et al., 2004), and (only) about 1 year around a cycle minimum (Legrand and Simon, 1981; Wang and Sheeley, 2009; Du, 2011b), as indicated in Fig. 10. The strength of geomagnetic activity can only be roughly evaluated from the strength of solar (sunspot) activity, as the linear correlation coefficient between the smoothed monthly mean $aa$ and $R_l$ is only 0.61 (Du, 2011c) or even lower (0.43) if using the non-smoothed series (Du, 2011b). In addition, the future solar activity is also unknown at the current time and so it can not be directly used to estimate the future geomagnetic activity.**

There are many methods that can be used to predict the maximum amplitude of sunspot cycle ($R_m$), such as 1) statistical methods, employing the relationship between the inter-cycle parameters (Thompson, 1988; Hathaway et al., 1994) or the early rising rate (Thompson, 1988; Cameron and Sch$\ddot{u}$ssler, 2008; Du and Wang, 2012); 2) the functional methods, using mathematical functions of a few parameters (Hathaway et al., 1994; Du, 2011d) for extrapolating the following monthly values; 3) the geomagnetic precursor methods (Ohl, 1979; Brown and Williams, 1969; Du et al., 2009),

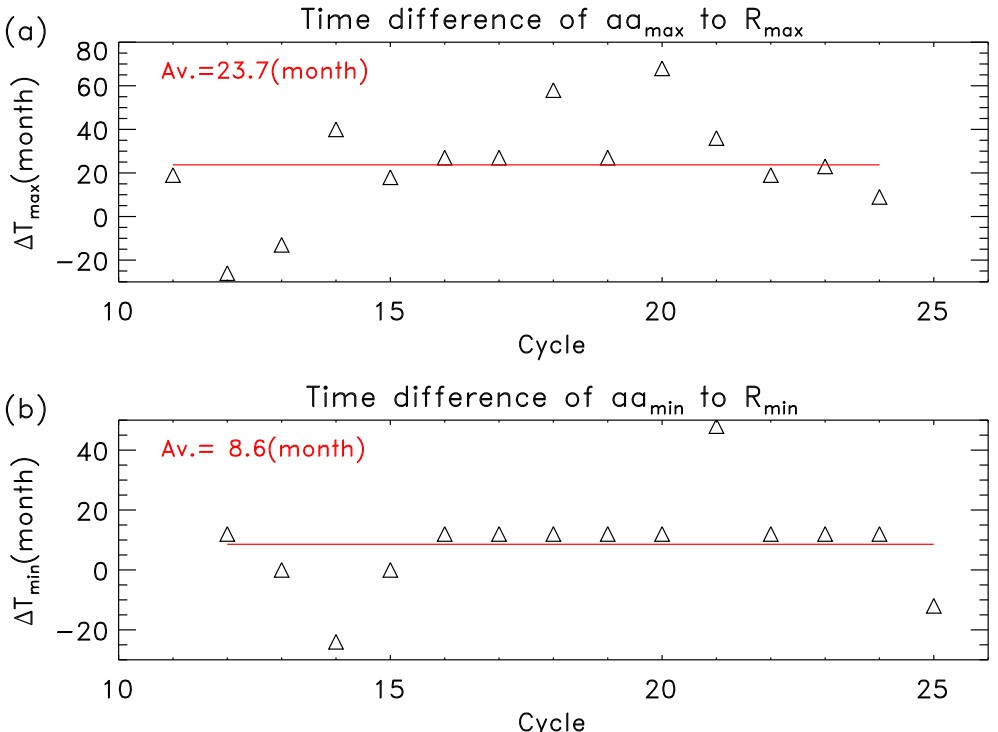

**Fig. 10.** The time difference between $aa_{max}$ and $R_{max}$ (a) and that between $aa_{min}$ and $R_{min}$ (b).

using the geomagnetic activity near the solar minimum; and 4) the solar precursor ones (Schatten et al., 1978; Pesnell and Schatten, 2018), using the previous cycle's polar field.

In contrast, there are less methods found to predict the maximum amplitude of ge-

omagnetic index for the 11-year solar cycle. Geomagnetic activity forecast has been over the order of hours or days (McPherron, 1999; Abunina et al., 2013). The annual or monthly prediction on the geomagnetic activity is within a limited accuracy (over 20%) due to the irregular variation in the time series (McPherron, 1999; Gordon, 2015). In the earlier years, Kane (1988) even pointed out that it is impossible to forecast the long-term geomagnetic activity through analyzing the time series of the $Ap$ and $aa$ indices (refer also to Gordon, 2015). The geomagnetic activity near the solar minimum or at the decreasing phase of the solar cycle has been widely used to predict the maximum amplitude of sunspot cycle, but was seldom used to predict the maximum amplitude of the geomagnetic activity itself.

In the current work, we analyzed the **highest ($aa_\mathsf{H}$) and the lowest ($aa_\mathsf{L}$) 3-hourly $aa$ index in each 3 days' interval, smoothed by 363 days (121 points) to mimic the 13 months smoothing. It is found that the maximum of $aa_\mathsf{H}$ ($aa_\mathsf{Hmax}$) is well correlated to the preceding minimum of either $aa_\mathsf{H}$ ($aa_\mathsf{Hmin}, r = 0.85$) or $aa_\mathsf{L}$ ($aa_\mathsf{Lmin}, r = 0.89$) for the 11-year solar cycle.** So, these relationships can be used to estimate the strength of geomagnetic activity for the ensuing cycle by employing the time series itself, $aa_\mathsf{Hmax}(25) = 84.5 \pm 6.9$ (nT). It implies that the strength of **geomagnetic activity for** the ensuing cycle 25 would be similar to the average over the past cycles, but higher than that of cycle 24 **by about 30%. Certainly, this estimate may be an upper limit, as cycle 24 has not completely passed and we should check if there is an even smaller value of $aa_\mathsf{Hmin}(25)$ or $aa_\mathsf{Lmin}(25)$ than that used in the current work (31.05 or 2.94) in the future few months.**

**Similar result can also be obtained if using the 363-day-smoothing highest/lowest 3-hourly $Ap$ index in 3-days-interval ($Ap_\mathsf{H}/Ap_\mathsf{L}$). The maximum of $Ap_\mathsf{H}$ ($Ap_\mathsf{Hmax}$) is found to be well correlated to the preceding minimum of $Ap_\mathsf{H}(Ap_\mathsf{Hmin}, r = 0.96)$ or $Ap_\mathsf{L}(Ap_\mathsf{Lmin}, r = 0.79)$ for the 11-year solar cycle. The rise time ($T_\mathsf{Ha}$) from $Ap_\mathsf{Hmin}$ to $Ap_\mathsf{Hmax}$ is reversely correlated to the preceding minimum of $Ap_\mathsf{L}(Ap_\mathsf{Lmin}, r = -0.72)$. For the 13-month smoothed monthly mean $aa(Ap)$ index, the maximum $aa(Ap)$ index, $aa_\mathsf{max}(Ap_\mathsf{max})$, of the solar cycle is also well correlated to the preceding minimum, $aa_\mathsf{min}(Ap_\mathsf{min})$, with a correlation coefficient of $r = 0.95(0.86)$.**

The well known 'Waldmeier effect' (Waldmeier, 1939) that the rise time of a solar cycle is well anti-correlated to the following maximum amplitude has been widely used to estimate the rise and peak times of a solar cycle if the amplitude has been predicted. However, such a correlation **is very weak** for the geomagnetic activity index. The rise time $(T_{Hr})$ from $aa_{Hmin}$ to $aa_{Hmax}$ for the 11-year solar cycle is found to be only weakly anti-correlated to the following maximum $(aa_{Hmax})$, $r = -0.42$. **The rise time $(T_{Ha})$ from $Ap_{Hmin}$ to $Ap_{Hmax}$ is also weakly anti-correlated to the following maximum $(Ap_{Hmax})$, $r = -0.33$. For the 13-month smoothed monthly mean $aa(Ap)$ index, the rise time of $aa_{max}(Ap_{max})$ is nearly uncorrelated to the following maximum, $r = -0.09(-0.17)$. These weak correlations may be related to the fact that the geomagnetic activity minimum (maximum) is not aligned to the solar (sunspot) activity minimum (maximum) in time, as shown in Fig. 10 for the time difference of $aa_{max}$ to $R_{max}$, $\Delta T_{max}$ (a), and that of $aa_{min}$ to $R_{min}$, $\Delta T_{min}$ (b). In most cases, $aa_{max}(aa_{min})$ lags behind $R_{max}(R_{min})$. But in some other cases, $aa_{max}(aa_{min})$ precedes $R_{max}(R_{min})$. The weak correlation between the rise time and the following maximum of geomagnetic activity for the 11-year solar cycle can hardly be used to estimate the former.**

According to the analysis above, the following conclusions may be summarized.

1. **The 363-day-smoothing highest $(aa_H)$ and lowest $(aa_L)$ 3-hourly $aa$ indices in 3-day-interval are analyzed, finding that the maximum of $aa_H$ $(aa_{Hmax})$ is well correlated to the preceding minimum of either $aa_H$ $(aa_{Hmin}, r = 0.85)$ or $aa_L$ $(aa_{Lmin}, r = 0.89)$ for the 11-year solar cycle.** As a result, the maximum $aa$ index for the ensuing cycle 25 is estimated to be $aa_{Hmax}(25) = 84.5 \pm 6.9$ (nT), about 30% higher than that of cycle 24. **This value is equivalent to the Ap index of $Ap_{Hmax}(25) = 47.8 \pm 3.9 \pm 2.1$ (nT) if using the relationship between $Ap$ and $aa$ (Eq.(9)).**

2. **The maximum $(aa_{Lmax})$ of $aa_L$ is also found to be well correlated to the preceding $aa_{Hmin}$, $r = 0.80$. Based on this correlation, $aa_{Lmax}(25)$ is estimated to be $6.1 \pm 1.2$ (nT), about 16% higher than that of cycle 24.**

3. **The maximum of sunspot cycle $(R_m)$ is much better correlated to the high geomag-**

netic activity ($aa_{\mathsf{Hmax}}$, $r = 0.79$) **than to the low one ($aa_{\mathsf{Lmax}}, r = 0.37$).**

4. **The rise time ($T_{\mathsf{Hr}}$) from $aa_{\mathsf{Hmin}}$ to $aa_{\mathsf{Hmax}}$ is found to be weakly anti-correlated to the following maximum ($aa_{\mathsf{Hmax}}$) for the 11-year solar cycle, $r = -0.42$ at the 84% confidence level.**

5. **Similar correlations are found for the 363-day-smoothing highest/lowest 3-hourly $Ap$ index in 3-day-interval ($Ap_{\mathsf{H}}/Ap_{\mathsf{L}}$). (1) The maximum of $Ap_{\mathsf{H}}$ ($Ap_{\mathsf{Hmax}}$) is well correlated to the preceding minimum of either $Ap_{\mathsf{H}}$ ($Ap_{\mathsf{Hmin}}, r = 0.96$) or $Ap_{\mathsf{L}}$ ($Ap_{\mathsf{Lmin}}, r = 0.79$) for the 11-year solar cycle. (2) The maximum of $Ap_{\mathsf{L}}$ ($Ap_{\mathsf{Lmax}}$) is well correlated to the preceding $Ap_{\mathsf{Hmin}}(r = 0.70)$. (3) The rise time ($T_{\mathsf{Ha}}$) from $Ap_{\mathsf{Hmin}}$ to $Ap_{\mathsf{Hmax}}$ is well anti-correlated to the preceding $Ap_{\mathsf{Lmin}}$ ($r = -0.72$).**

6. **For the 13-month smoothed monthly mean $aa(Ap)$ index, the maximum $aa(Ap)$ index, $aa_{\mathsf{max}}(Ap_{\mathsf{max}})$, of the solar cycle is well correlated to the preceding minimum, $aa_{\mathsf{min}}(Ap_{\mathsf{min}})$, with a correlation coefficient of $r = 0.95(0.86)$. The rise time of $aa_{\mathsf{max}}(Ap_{\mathsf{max}})$ is nearly uncorrelated to the following maximum, $r = -0.09(-0.17)$.**

*Acknowledgements.* We are grateful to the two anonymous referees for valuable suggestions which improved this manuscript. This work is supported by the National Science Foundation of China (NSFC) through grants 11973058 and 11603040.

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
