# Peer review of "Estimating the maximum of smoothed highest 3-hourly *aa* index in 3 days by the preceding minimum for the solar cycle"

_Annales Geophysicae, 2020_

## Referee Comment (RC1) · Anonymous Referee #1 · 28 Apr 2020

The author uses smoothed monthly aa/Ap index to study the relation between the minimum and maximum aa/Ap values in order to predict the maximum value of the aa/Ap index.

Usually long-term smoothing is used to study the solar cycle; to show the correlation between the solar cycle and the index variations.

Due to the small number of high amplitude values, smoothing removes all the high amplitude maxima and move the data towards the minimum.

The author states that "the maximum aa index for the ensuing cycle 25 is predicted to be aamax(25) = 26.9±2.6." This is very small value and it could be mistakenly understood that this solar cycle will be very quiet.

[Figure]

The values listed in Table 1 under the aamax are much smaller than those observed in any disturbed day. These values can't represent the maximum aa index or the strength of the geomagnetic activities. As it could be seen from Fig. 1 the aa index has arrive to a peak value of about 67 nT in 19 March 2020 and the Kp value for this time is 4+

Also the paper is based on the data listed in Table 1. Which have been retrieved from smoothed aa index data. The smoothing could be done in many different ways each will produce different data sets.

However, when considering the geomagnetic activities, we are usually interested to know how sever it will be and for how long it will last.

Therefore, I suggest the following It should be stated clearly that these max values are for smoothed aa index and it should be given a special note. The paper title should also indicate this.

The author could try to compare the expect strength of the 25 cycle with the previous cycles. So, we could understand is it will be more active or less active.

The author could try to predict a more reliable maximum of the aa index for the 25th cycle. To do so I could suggest to construct two data sets of the observed aa index minimum and maximum values for each 3 days or more. These two sets could be smoothed for 13 months. The correlation between these two data sets (for 3 days min and max values) are about 0.79 From these two data sets the author could peak the maximum and minimum aa index for each solar cycle and replace these values with those in Table 1.

Finally, the units of the indices (nT) should be written in text and on the Figures.
* * *
[Figure]

**DATA PLOT**

**Plot of *aa* index from 2020-03-18 to 2020-03-20**

2020-03-19 01:30
━Provisional aa: 67
━Provisional Kpa: 4+

Return to data plot

**Fig. 1.** aa index for the period 18-20 March 2020

**Fig. 2.** Smoothed Max aa index

**Fig. 3.** Smoothed Min aa index

[Figure]

---

## Author Comment (AC1) · 13 May 2020

1) Therefore, I suggest the following It should be stated clearly that these max values are for smoothed aa index and it should be given a special note. The paper title should also indicate this.

R: Yes. Thank you.

2) The author could try to compare the expect strength of the 25 cycle with the previous cycles. So, we could understand is it will be more active or less active.

R: Yes. We do.

3) The author could try to predict a more reliable maximum of the aa index for the 25th

cycle. To do so I could suggest to construct two data sets of the observed aa index minimum and maximum values for each 3 days or more. These two sets could be smoothed for 13 months. The correlation between these two data sets (for 3 days min and max values) are about 0.79 From these two data sets the author could peak the maximum and minimum aa index for each solar cycle and replace these values with those in Table 1.

R: Yes. We did as suggested. One paragraph is inserted in Section 4 (Discussions) to discuss the correlation between the smoothed monthly mean aa and RI. In the revised manuscript, we used only the 3-hourly aa index of ISGI since 1868. For each 3-day-interval, we find the highest aa index (aaH) and the lowest aa index (aaL) from 24 values of the 3-hourly aa indices. Then, both aaH and aaL are smoothed by 363 days (121 points) to mimic the 13-month smoothing. The results are similar to those using 13-month smoothed monthly mean values, apart from that the maximum is estimated to be around 85 for the highest value. The Ap index is used only for comparison.

4) Finally, the units of the indices (nT) should be written in text and on the Figures.

R: Yes. We do.

Please also note the supplement to this comment:
https://www.ann-geophys-discuss.net/angeo-2020-15/angeo-2020-15-AC1-supplement.pdf

**Supplement:**

Manuscript prepared for J. Name
with version 2.2 of the LATEX class copernicus‗discussions.cls.
Date: 13 May 2020

**Predicting the maximum of 363-day smoothed highest 3-hourly $aa$ index in 3-day-interval with the preceding minimum of 11-year cycle**

**Zhanle Du**

Key Laboratory of Solar Activity, National Astronomical Observatories, Chinese Academy of Sciences, Beijing 100012, China.

Correspondence to: Z. L. Du
(zldu@nao.cas.cn)

**Abstract**

Predicting the strength and peak time of geomagnetic activity for the ensuing cycle 25 is important in space weather service for planning future space missions. This study analyzed the **highest ($aa_H$) and lowest $aa$ index ($aa_L$) from 24 three-hourly values in 3-day-interval, smoothed by 363 days (121 points) to mimic the 13-month smoothing. It is found that the maximum ($aa_{Hmax}$) of $aa_H$ is well correlated to both the preceding minimum ($aa_{Hmin}, r = 0.85$) of $aa_H$ and the preceding minimum ($aa_{Lmin}, r = 0.89$) of $aa_L$ for the 11-year solar cycle. Based on these relationships the strength of geomagnetic activity for the ensuing cycle is predicted to be $aa_{Hmax}(25) = 85.5 \pm 6.9$ (nT), about 32% stronger than that of cycle 24. This value is equivalent to the Ap index of $Ap_{max}(25) = 56.0 \pm 4.8 \pm 1.2$ (nT). The maximum ($aa_{Lmax}$) of $aa_L$ is also found to be well correlated to the preceding $aa_{Hmin}(r = 0.80)$. The solar activity is much better correlated to the strong geomagnetic activity ($aa_{Hmax}, r = 0.79$) than to the weak one** ($aa_{Lmax}, r = 0.37$). The rise time of $aa_{Hmax}$ ($T_{Hr}$) is found to be weakly anti-correlated to the following maximum ($aa_{Hmax}$), $r = -0.42$ at the 84% confidence level. **Using this correlation and the predicted $aa_{Hmax}(25)$, one can roughly estimate the rise time, $T_{Hr}(25) = 5.2 \pm 2.0$ (years)**, implying that the geomagnetic activity for the ensuing cycle 25 would peak around August 2025$\pm$2.0 (years).

**1  Introduction**

Studying and predicting geomagnetic activities are important in both geophysics and space weather. Severe geomagnetic activities may cause intense geomagnetic storms (Gonzalez et al., 1989, 1994; Chen et al., 2019), leading to disruptions in communication and deviations of spacecrafts. With the current solar cycle 24 approaching its end, satellite and spacecraft-related departments want to know the strengths of both solar and geomagnetic activities in the ensuing cycle 25 for planning future space missions.

Among various indices to quantitatively describe the geomagnetic activity, the $aa$ index (Mayaud, 1972), derived from the 3-hourly K indices at two near-antipodal midlatitude stations in England and Australia, is the longest time series (since 1868) and has been widely used for analyzing long-term trends in the global geomagnetic activity (Russell and Mulligan, 1995; Marat et al., 2017; Du, 2011a; El-Borie et al., 2019) and for analyzing its correlation with both climate change (Cliver et al., 1998; Dobrica et al., 2009; Gavrilyeva et al., 2017) and solar activity (Echer et al., 2004; Prestes et al., 2006; Lukianova et al., 2009; Du, 2011b,c; Du and Wang, 2012; Singh and et al., 2019). The minimum $aa$ index ($aa_{\mathrm{min}}$), at or near the minimum of the solar cycle, has been widely used in predicting the maximum amplitude of the sunspot cycle ($R_{\mathrm{m}}$), the so-called Ohl's precursor method (Ohl, 1979; Brown and Williams, 1969; Du et al., 2009). But it is seldom used to directly predict the maximum $aa$ index ($aa_{\mathrm{max}}$) of an ensuing cycle.

The planetary geomagnetic index $Ap$ (Bartels, 1963) available since 1932, derived from the average of the measurements at 13 observatories around the globe, is a daily measure of the response of geomagnetic field to variations in the interplanetary magnetic field (IMF) and the solar wind (Li, 1997; McPherron, 1999; Tsurutani et al., 2006). It is the main global magnetic index forecasted by government agencies (McPherron, 1999). Most works on forecasting geomagnetic activity have been over short intervals, on the order of hours or days (McPherron, 1999; Abunina et al., 2013). In the earlier years, Kane (1988) pointed out that it is impossible to forecast the long-term geomagnetic activity through analyzing the daily, monthly and annual values of $Ap$ and $aa$ indices. Gordon (2015) demonstrated that long-term geomagnetic activity can only be predicted to within a limited threshold of accuracy due to the irregular trends and cycles in the annual data and nonlinear variability in the monthly series, through analyzing the $aa$ index.

In this study, we analyze the **highest ($aa_{\mathsf{H}}$) and lowest ($aa_{\mathsf{L}}$) 3-hourly $aa$ index in each 3 days' interval, smoothed by 363 days (121 points) to mimic the 13-month smoothing. It is found that the maximum ($aa_{\mathsf{Hmax}}$) of $aa_{\mathsf{H}}$ is well correlated to both the preceding minimum ($aa_{\mathsf{Hmin}}$) of $aa_{\mathsf{H}}$ and the preceding minimum ($aa_{\mathsf{Lmin}}$) of $aa_{\mathsf{L}}$ for the 11-year solar cycle, which can be used to predict $aa_{\mathsf{Hmax}}$.**

This study is arranged as follows. The data used in the current work are shown

in Sect. 2. Section 3 is devoted for the results. First, in Sect. 3.1, we **analyze** the relationships between the maximum **of the smoothed highest 3-hourly** $aa$ **index** ($aa_{Hmax}$) **in 3 days and the preceding minima of both the smoothed highest** ($aa_{Hmin}$) **and lowest** ($aa_{Lmin}$) **3-hourly** $aa$ **indices in 3 days, followed by a prediction of** $aa_{Hmax}$ for cycle 25.

5   **The relationship between the maximum of** $aa_L$ ($aa_{Lmax}$) **and the preceding** $aa_{Hmin}$ **is simply analyzed in Sect. 3.2.** In Sect. 3.3, we analyze the relationship between the rise time of $aa_H$ from $aa_{Hmin}$ to $aa_{Hmax}$ and the following maximum for the 11-year cycle, so as to estimate the peak time of geomagnetic activity for the ensuing cycle. Some conclusions are discussed and summarized in Sect. 4.

10 ## 2   Data

In this study, we use the **3-hourly** $aa$ index since 1868 from the International Service of Geomagnetic Indices (ISGI)[1]. **For each 3-day-interval, we find the highest** $aa$ **index** ($aa_H$) **and the lowest** $aa$ **index** ($aa_L$) **from 24 values of the 3-hourly** $aa$ **indices. Then, both** $aa_H$ **and** $aa_L$ **are smoothed by 363 days (121 points) to mimic the 13-month smoothing,**
15 **as shown in Fig. 1 (solid). The (13-month) smoothed monthly mean International sunspot number series (** $R_I$ **, Clette et al., 2016) of the second version[2] is used for comparison (dotted).**

  The parameters used in this study are listed in Table 1, in which $aa_{Hmin}$ ($aa_{Hmax}$) is the minimum (maximum) of $aa_H$, $aa_{Lmin}$ ($aa_{Lmax}$) the minimum (maximum) of $aa_L$, $T_{Hr}$
20 the rise time of $aa_H$ from $aa_{Hmin}$ to $aa_{Hmax}$, and $R_m$ the maximum of $R_I$ for the 11-year solar cycle. The last row denotes the averages of the parameters.
* * *
[1]http://isgi.unistra.fr/
[2]http://www.sidc.be/silso/datafiles

**Table 1.** The minimum ($aa_{Hmin}$) and maximum ($aa_{Hmax}$) of the 363-day smoothed highest 3-hourly $aa$ index ($aa_H$) in each 3 days, the rise time of $aa_{Hmax}$ ($T_{Hr}$), the minimum ($aa_{Lmin}$) and maximum ($aa_{Lmax}$) of the 363-day smoothed lowest 3-hourly $aa$ index ($aa_L$) in each 3 days, and the maximum ($R_m$) of 13-month smoothed sunspot number for solar cycles 11-25.

| $n$ | $aa_{Hmin}$(nT) | $aa_{Hmax}$(nT) | $T_{Hr}$(yr) | $aa_{Lmin}$(nT) | $aa_{Lmax}$(nT) | $R_m$ |
|---|---|---|---|---|---|---|
| 11 | | 88.24 | | | 4.29 | 234.0 |
| 12 | 21.41 | 79.60 | 3.81 | 2.00 | 4.60 | 124.4 |
| 13 | 31.88 | 86.92 | 2.08 | 2.58 | 4.47 | 146.5 |
| 14 | 19.40 | 59.55 | 8.47 | 2.00 | 4.37 | 107.1 |
| 15 | 25.31 | 73.28 | 5.72 | 2.19 | 5.80 | 175.7 |
| 16 | 30.80 | 75.90 | 5.56 | 2.07 | 7.56 | 130.2 |
| 17 | 41.60 | 84.01 | 4.01 | 2.52 | 6.36 | 198.6 |
| 18 | 51.13 | 88.31 | 6.57 | 3.02 | 7.18 | 218.7 |
| 19 | 49.60 | 100.99 | 5.89 | 3.08 | 7.66 | 285.0 |
| 20 | 43.08 | 79.31 | 7.54 | 2.55 | 8.79 | 156.6 |
| 21 | 49.35 | 97.87 | 2.63 | 3.52 | 7.96 | 232.9 |
| 22 | 48.93 | 103.89 | 4.78 | 3.57 | 9.73 | 212.5 |
| 23 | 47.84 | 99.75 | 5.92 | 2.98 | 10.22 | 180.3 |
| 24 | 25.02 | 64.81 | 5.73 | 2.17 | 5.27 | 116.4 |
| 25 | 33.24 | | | 2.94 | | |
| Av. | 37.04 | 84.46 | 5.29 | 2.66 | 6.73 | 179.9 |

[Figure]

**Fig. 1.** (a) The highest $aa$ index ($aa_H$) and (b) the lowest $aa$ index ($aa_L$) in each 3 days, smoothed by 363 days. The numbers in the figure indicate the 11-year solar cycles. The upper dashed and lower dash-dotted lines indicate the maxima and minima, respectively, for the 11-year cycle. The dotted line represents the smoothed monthly mean sunspot number ($R_I$) for comparison.

**3  Result**

The correlation coefficients between the parameters in Table 1 are listed in Table 2 for comparison. It is seen in Table 2 that $R_m$ is well correlated to $aa_{Hmin}(r = 0.84)$, $aa_{Hmax}(r = 0.79)$, $aa_{Lmin}(r = 0.81)$, and positive correlated to $aa_{Lmax}(r = 0.37)$. It implies that the stronger the solar activity ($R_I$), the higher the geomagnetic activity ($aa$). But the solar activity is much better correlated to the strong geomagnetic activity ($aa_{Hmax}$) than to the weak one ($aa_{Lmax}$), the latter may be due to the high speed solar wind streams which is out of phase with the solar activity cycle (Feynman, 1982; Hathaway and Wilson, 2006). The

**Table 2.** The correlation coefficient ($r$) between parameters $x$ and $y$.

| $x/y$ | $aa_{Hmin}$ | $aa_{Hmax}$ | $T_{Hr}$ | $aa_{Lmin}$ | $aa_{Lmax}$ | $R_m$ |
|---|---|---|---|---|---|---|
| $aa_{Hmin}$ | 1.00 | 0.85 | $-0.10$ | 0.87 | 0.80 | 0.84 |
| $aa_{Hmax}$ | 0.85 | 1.00 | $-0.42$ | 0.89 | 0.63 | 0.79 |
| $T_{Hr}$ | $-0.10$ | $-0.42$ | 1.00 | $-0.28$ | 0.13 | $-0.18$ |
| $aa_{Lmin}$ | 0.87 | 0.89 | $-0.28$ | 1.00 | 0.70 | 0.81 |
| $aa_{Lmax}$ | 0.80 | 0.63 | 0.13 | 0.70 | 1.00 | 0.37 |

correlation between $R_m$ and $aa_{Hmin}$ (or $aa_{Lmin}$) is related to the Ohl's precursor method (Ohl, 1979) for predicting $R_m$. Other significant correlations can be used to predict $aa_{Hmax}$ (Sect. 3.1), $aa_{Lmax}$ (Sect. 3.2), and $T_{Hr}$ (Sect. 3.3) in this study.

**3.1 Relationship between the maximum and preceding minimum**

5 It is seen in Table 2 that $aa_{Hmax}$ is well correlated to the preceding $aa_{Hmin}(r=0.85)$ and $aa_{Lmin}(r=0.89)$, as shown in Fig. 2 for the scatter plots of $aa_{Hmax}$ against $aa_{Hmin}$(a) and $aa_{Lmin}$(b). The solid line represents a linear fit of $aa_{Hmax}$ to $aa_{Hmin}$ ($aa_{Lmin}$) with the least-squares-fit regression equations given by

$$\begin{cases} aa_{Hmax} = 47.1 \pm 7.1 + (0.99 \pm 0.18) aa_{Hmin}, \sigma = 7.3, \\ aa_{Hmax} = 25.3 \pm 9.4 + (22.3 \pm \quad 3.5) aa_{Lmin}, \sigma = 6.5, \end{cases} \tag{1}$$

10 where $\pm$ represents the $1\sigma$ deviation and $\sigma$ the standard deviation of the fitting.

Therefore, the $aa$ index at the minimum can be used as an indicator to predict the index at the maximum. From the above relationships, one can predict $aa_{Hmax}$ for cycle $n=25$ by substituting the values of $aa_{Hmin}$(33.24 nT) and $aa_{Lmin}$(2.94 nT) into these equations,

$$\begin{cases} aa_{Hmax1}(25) = 80.1 \pm 7.3 (\text{nT}), \text{ from } aa_{Hmin}, \\ aa_{Hmax2}(25) = 91.0 \pm 6.5 (\text{nT}), \text{ from } aa_{Lmin}, \end{cases} \tag{2}$$

[Figure]

**Fig. 2.** (a) Scatter plot of $aa_{\mathrm{Hmax}}$ against $aa_{\mathrm{Hmin}}$ (triangles) and the linear fit (solid). (b) Scatter plot of $aa_{\mathrm{Hmax}}$ against $aa_{\mathrm{Lmin}}$ (triangles) and the linear fit (solid).

**(labelled by asterisk). As the above relationships have a similar correlation (0.85 and 0.89), we take the prediction of $aa_{\mathrm{Hmax}}(25)$ as the average,**

$$aa_{\mathrm{Hmax}}(25) = \tfrac{1}{2}[aa_{\mathrm{Hmax1}}(25) + aa_{\mathrm{Hmax2}}(25)]$$
$$= 85.5 \pm 6.9 (\mathrm{nT}).$$

(3)

**It implies that the 363-day smoothed highest 3-hourly $aa$ index in 3-day-interval during the maximum period of cycle 25 is predicted to be close to the average (84.46 nT) over the past cycles (Table 1), but higher than that (64.81 nT) of cycle 24 by about 32.0%.**

**It should be pointed out that the above prediction may be an upper estimate for the maximum $aa$ index as cycle 24 has not completely passed. Although we are not quite sure if the current $aa_{\mathrm{H}}$ ($aa_{\mathrm{L}}$), 33.94 (3.47) in November 2019, would decrease to a smaller value than that, 33.24 (2.94), used in the current work, there would not be significant variations**

in $aa_{Hmin}$, $aa_{Lmiin}$ and the above prediction, as the solar (sunspot) activity ($R_I$) shows a sign to stop decreasing and to oscillate around the minimum in the recent few months.

**3.2 Relationship between $aa_{Lmax}$ and the preceding $aa_{Hmin}$**

It is seen in Table 2 that $aa_{Lmax}$ is also well correlated to the preceding $aa_{Hmin}$ ($r = 0.80$), as shown in Fig. 3(a) for the scatter plot of $aa_{Lmax}$ against $aa_{Hmin}$. The linear fitting equation of $aa_{Lmax}$ to $aa_{Hmin}$ (solid) is

$$aa_{Lmax} = 2.0 \pm 1.2 + (0.131 \pm 0.030)aa_{Hmin}, \ \sigma = 1.2. \tag{4}$$

Substituting $aa_{Hmin}(25) = 33.24$ (nT) into this equation, one can estimate the 363-day smoothed lowest 3-hourly $aa$ index in 3-day-interval during the maximum period of cycle 25, $aa_{Lmax} = 6.4 \pm 1.2$ (nT). This value is close to the average (6.73 nT) over the past cycles, but higher than that (5.27 nT) of cycle 24 by 21.1%.

**3.3 Relationship between the rise time and maximum**

At last, in this section, we analyze if the rise time of the geomagnetic index for the 11-year cycle is correlated to the following maximum so that it can be used to estimate the rise time, as the case in sunspot cycle (Waldmeier, 1939).

Figure 3(b) shows the scatter plot of the rise time ($T_{Hr}$) from $aa_{Hmin}$ to $aa_{Hmax}$ for the 11-year cycle against its following maximum ($aa_{Hmax}$). It is seen in this figure that $T_{Hr}$ is weakly anti-correlated to $aa_{Hmax}$, with a correlation coefficient of $r = -0.42$ at a confidence level of about 84%. The linear fitting equation of $T_{Hr}$ to $aa_{Hmax}$ is

$$T_{Hr} = 9.9 \pm 3.0 - (0.055 \pm 0.036)aa_{Hmax}, \ \sigma = 1.6. \tag{5}$$

Using this weak correlation, one can roughly estimate the rise time, $T_{Hr}(25) = 5.2 \pm 0.4 \pm 1.6$ (years), by substituting the predicted value of $aa_{Hmax}(25) = 85.5 \pm 6.9$ (nT) in Eq. (3) into this equation, here $\pm 0.4$ is derived from the uncertainty ($\pm 6.9$) of $aa_{Hmax}(25)$. Certainly, this estimate is less reliable than the prediction on the maximum.

[Figure]

**Fig. 3.** (a) Scatter plot of $aa_{Lmax}$ against $aa_{Hmin}$ (triangles) and the linear fit (solid). (b) Scatter plot of $T_{Hr}$ against $aa_{Hmax}$ (triangles) and the linear fit (solid).

At the current state, the smoothed monthly mean sunspot number is as low as **3.1 in September 2019**. The monthly mean values for the last few months appear in an oscillation around the minimum. So the minimum time of cycle 25 should not be far from September 2019 ($\pm$ **8 months, Du and Wang, 2011**). If the minimum time of $aa$ index is (temporarily) taken as that of sunspot activity with an average delay of **9 months (Legrand and Simon, 1981; Wang and Sheeley, 2009; Du, 2011b)**, then the peak time of $aa_{Hmax}(25)$ would be roughly estimated to be around September 2019 $+9$ (months) $+T_{Hr}(25) \sim$ **August 2025**$\pm 2.0$ **(years).**

**4 Discussions and Conclusions**

It is well known that the aa index is positively correlated to the solar activity ($R_l$). In general, the stronger the solar activity, the higher the ($aa$) geomagnetic activity. But the correlation between $aa$ and $R_l$ is not very strong (Du, 2011b,c). We can only roughly evaluate the strength of geomagnetic activity from the strength of solar (sunspot) activity. In addition, the future solar activity is also unknown at the present time. The relationship between $aa$ and $R_l$ is not a simple linear one (Borello-Filisetti et al., 1992; Mussino et al., 1994; Kishcha et al., 1999; Lockwood et al., 1999; Echer et al., 2004). The $aa$ index tends to lag behind $R_l$ about 2–3 years around a solar cycle maximum (Wang et al., 2000; Echer et al., 2004), and (only) about 1 year around a cycle minimum (Legrand and Simon, 1981; Wang and Sheeley, 2009; Du, 2011b). The complicate relationship between $aa$ and $R_l$ can be better understood by an integral response model (Du, 2011c) that increases the correlation from 0.61 to 0.85.

There are many methods that can be used to predict the maximum amplitude of sunspot cycle ($R_m$), such as 1) statistical methods, employing the relationship between the inter-cycle parameters (Thompson, 1988; Hathaway et al., 1994) or the early rising rate (Thompson, 1988; Cameron and Sch$\ddot{u}$ssler, 2008; Du and Wang, 2012); 2) the functional methods, using mathematical functions of a few parameters (Hathaway et al., 1994; Du, 2011d) for extrapolating the following monthly values; 3) the geomagnetic precursor methods (Ohl, 1979; Brown and Williams, 1969; Du et al., 2009), using the geomagnetic activity near the solar minimum; and 4) the solar precursor ones (Schatten et al., 1978; Pesnell and Schatten, 2018), using the previous cycle's polar field.

In contrast, there are less methods found to predict the maximum amplitude of geomagnetic index. Geomagnetic activity forecast has been over the order of hours or days (McPherron, 1999; Abunina et al., 2013). The annual or monthly prediction on the geomagnetic activity is within a limited accuracy (over 20%) due to the irregular variation in the time series (McPherron, 1999; Gordon, 2015). In the earlier years, Kane

(1988) even pointed out that it is impossible to forecast the long-term geomagnetic activity through analyzing the time series of the $Ap$ and $aa$ indices (refer also to Gordon, 2015). The geomagnetic activity near the solar minimum or at the decreasing phase of the solar cycle has been widely used to predict the maximum amplitude of sunspot cycle, but were seldom used to predict the maximum amplitude of the geomagnetic activity itself.

In the current work, we analyzed the **highest ($aa_H$) and the lowest ($aa_L$) 3-hourly $aa$ index in each 3-day-interval, smoothed by 363 days (121 points) to mimic the 13 months smoothing. It is found that the maximum ($aa_{Hmax}$) of $aa_H$ is well correlated to both the preceding minimum ($aa_{Hmin}, r = 0.85$) of $aa_H$ and the preceding minimum ($aa_{Lmin}, r = 0.89$) of $aa_L$ for the 11-year solar cycle.** So, these relationships can be used to predict the strength of geomagnetic activity for the ensuing cycle, $aa_{Hmax}(25) = 80.1 \pm 7.3$ (nT) or $91.0 \pm 6.5$ (nT), with an average of $aa_{Hmax}(25) = 85.5 \pm 6.9$ (nT). It implies that the strength of **geomagnetic activity for** the ensuing cycle 25 would be similar to the average over the past cycles, but higher than that of cycle 24 **by about 32%. Certainly, this value is an upper estimate, as cycle 24 has not completely passed and we should check if there is an even smaller value of $aa_{Hmin}(25)$ or $aa_{Lmin}(25)$ than that used in the current work (33.24 or 2.94) in the future few months.**

**If using the high correlation between the smoothed monthly mean $Ap$[3] and $aa$ indices since 1932, $r = 0.94$, and the linear fitting equation of $Ap$ to $aa$,**

$$Ap = -1.35 \pm 0.17 + (0.694 \pm 0.008)aa, \sigma = 1.2, \tag{6}$$

**the above prediction will be equivalent to $Ap_{max}(25) = 56.0 \pm 4.8 \pm 1.2$ (nT), here $\pm 4.8$ is derived from the uncertainty ($\pm 6.9$) of $aa_{Hmax}(25)$.**

The well known 'Waldmeier effect' (Waldmeier, 1939) that the rise time of a solar cycle is well anti-correlated to the amplitude has been widely used to estimate the rise and peak times of a solar cycle if the amplitude has been predicted. However, such a correlation **is very weak** in the $aa$ geomagnetic index. The rise time of $aa_{Hmax}(T_{Hr})$
* * *
[3]http://www.gfz-potsdam.de/en/kp-index

for the 11-year cycle is found to be weakly anti-correlated to the following maximum ($aa_{\mathrm{Hmax}}$), $r = -0.42$ **at the 84% confidence level.** Using this correlation, one could roughly estimate the rise time, $T_{\mathrm{Hr}}(25) = 5.2 \pm 2.0$ (years), and the peak time, August 2025$\pm$2.0 (years), of geomagnetic activity for the ensuing cycle 25. Certainly, this estimate is much less reliable than the predictions on the maximum.

According the analysis above, the following conclusions may be summarized,

1. **The maximum ($aa_{\mathrm{Hmax}}$) of the 363-day smoothed highest 3-hourly** $aa$ **index in 3-day-interval ($aa_{\mathrm{H}}$) is found to be well correlated to both the preceding minimum ($aa_{\mathrm{Hmin}}, r = 0.85$) of** $aa_{\mathrm{H}}$ **and the preceding minimum ($aa_{\mathrm{Lmin}}, r = 0.89$) of the 363-day smoothed lowest 3-hourly** $aa$ **index in 3-day-interval ($aa_{\mathrm{L}}$) for the 11-year solar cycle.** As a result, the maximum for the ensuing cycle 25 is predicted to be $aa_{\mathrm{Hmax}}(25) = 85.5 \pm 6.9$ (nT), about 32% higher than that of cycle 24. **This value is equivalent to the Ap index of** $Ap_{\mathrm{max}}(25) = 56.0 \pm 4.8 \pm 1.2$ **(nT).**

2. **The maximum ($aa_{\mathrm{Lmax}}$) of** $aa_{\mathrm{L}}$ **is also found to be well correlated to the preceding** $aa_{\mathrm{Hmin}}$**,** $r = 0.80$**. Based this correlation,** $aa_{\mathrm{Lmax}}(25)$ **is predicted to be** $6.4 \pm 1.2$ **(nT).**

3. **The solar activity is much better correlated to the strong geomagnetic activity ($aa_{\mathrm{Hmax}}$,** $r = 0.79$**) than to the weak one ($aa_{\mathrm{Lmax}}, r = 0.37$).**

4. **The rise time ($T_{\mathrm{Hr}}$) is found to be weakly correlated to the following maximum ($aa_{\mathrm{Hmax}}$) for the 11-year cycle,** $r = -0.42$ **at the 84% confidence level.** Using this correlation, one could roughly estimate the rise time, $T_{\mathrm{Hr}}(25) = 5.2 \pm 2.0$ (years), and the peak time, August 2025$\pm$2.0 (years), of geomagnetic activity for the ensuing cycle 25.

*Acknowledgements.* We are grateful to the anonymous referee for valuable suggestions which improved this manuscript. This work is supported by the National Science Foundation of China (NSFC) through grants 11973058 and 11603040.

[revised manuscript text omitted]

---

## Referee Comment (RC2) · Anonymous Referee #2 · 25 Jun 2020

The author identified maximum and minimums of smoothed aa and Ap indices through several solar cycles and calculate correlations between minimum and maximum values and between min/max values with respect to the preceding-following cycle. The relations that are found through the means of linear regression are then use to predict estimated aa/Ap minimum/maximum values for solar cycle 25.

Main comments

1. My main concern is with the selection of the dataset. It is not clear to me why the author choose to work with the 13-month smoothed aa index instead of the highest resolution available. Smoothing everything will naturally result in predictions that converge to the mean values and therefore fail to capture the spiky behavior of storm indices. This is particularly relevant in the case of Ap index. As shown in Table 1,
the Ap smoothed monthly means corresponds to period of at most minor geomagnetic activity. Therefore all storm activity is lost. I suggest the author repeat the calculations using the highest available temporal resolution of the indices and compare them with the current results of the manuscript.

2 Page 2 L19-21 These results are hardly relevant. Simpler methods will estimate the duration of solar cycle phases with significantly better accuracy (For example, NOAA predicts a rising duration with an error of $\sim$8 months). Estimating the duration of half a cycle with an uncertainty of almost half the solar cycle results in a disconnection between the mathematical results and the known repetitiveness of the studies phenomena. There's a reason it is called the 11-year cycle. I suggest the author to revise the calculations and to interpret them in the context of what could be a reasonable assumption of the duration of the phases of SC25.

3. The main results of the paper (shown in Figures 1-4) are heavily influenced by the decision of using smoothed indices. While they may be correct in that particular context, the author should consider if the methodology utilized is the appropriate for this particular problem. Going back to point 1, if a different dataset is utilized, all figures need to be remade. On a note regarding presentation of the figures, adding colors to the different lines and making the figures of the appropriate size will significantly improve the readability.

Specific comments

Title: I suggest replacing "minimum" with "solar minimum" to explicitly refer to solar cycle. Note that currently the title is misleading, as the prediction is regarding the smoothed data. Please adjust accordingly.

L10 - What is the meaning of a double plus-minus. Is it referring to different error sources? In that case please specify. L18 - Do you mean anti-correlated? L26 What

do you mean by deviations? Please provide relevant references.

L2-3 Predicted or estimated? A prediction is a statement about the future. A correlation between two variables at most indicates the ability to estimate one when the other is available, which appear to be the case.

L2-4 This extremely high correlation is clearly affected by the process of smoothing the data. Similar with other figures and equations, please correct based on major comments.

---

## Author Comment (AC2) · 19 Jul 2020

Reply to Referee # 1 Overall modifications The manuscript has been thoroughly revised based on two referees. We discuss mainly the result using the 3-hourly aa index since 1868 in Sect 2. For each 3 days' interval, we find out the highest/lowest aa index (aaH/aaL) from 24 values of the 3-hourly aa indices. In order to reduce accidental events in the data, both aaH and aaL are smoothed by 363 days (121 points) to mimic the 13-month smoothing, as suggested by Referee 1. The maximum of aaH (aaHmax) is found to be well correlated to the preceding minimum of either aaH (aaHmin, r=0.85) or aaL (aaLmin, r=0.89) for the 11-year solar cycle. Based on these correlations, the strength of geomagnetic activity for cycle 25 is estimated to be aaHmax (25)= 85.5±6.9 (nT), similar to the average over the past cycles, but about 32% higher than that of

cycle 24. The rise time (THr) from aaHmin to aaHmax is found to be only weakly anti-correlated to the following aaHmax, r=-0.42. Such a weak correlation is no longer used to estimate THr as suggested by Referee 2. Similar result can also be obtained if using the 363-day-smoothing highest/lowest 3-hourly Ap index in 3-day-interval (ApH/ApL), shown in Sect. 3. The maximum of ApH (ApHmax) is well correlated to the preceding minimum of ApH (ApHmin, r=0.96) or ApL (ApLmin, r=0.79) for the 11-year solar cycle. The rise time (THa) from ApHmin to ApHmax is reversely correlated to the preceding minimum of ApL (ApLmin, r=-0.72). For the 13-month smoothed monthly mean aa (Ap) index, the result is moved down to Sect. 4, retained as a comparison, as suggested by Referee 2, but using only the aa index since 1868. The maximum aa(Ap) index, aamax (Apmax), of the solar cycle is also well correlated to the preceding minimum, aamin ( Apmin), with a correlation coefficient of r= 0.95(0.86). 'Predict' is changed to 'estimate' as suggested by Referee 2. ——————— The author uses smoothed monthly aa/Ap index to study the relation between the minimum and maximum aa/Ap values in order to predict the maximum value of the aa/Ap index. Usually long-term smoothing is used to study the solar cycle; to show the correlation between the solar cycle and the index variations. Due to the small number of high amplitude values, smoothing removes all the high amplitude maxima and move the data towards the minimum. The author states that "the maximum aa index for the ensuing cycle 25 is predicted to be aamax(25) = 26.9±2.6." This is very small value and it could be mistakenly understood that this solar cycle will be very quiet. The values listed in Table 1 under the aamax are much smaller than those observed in any disturbed day. These values can't represent the maximum aa index or the strength of the geomagnetic activities. As it could be seen from Fig. 1 the aa index has arrive to a peak value of about 67 nT in 19 March 2020 and the Kp value for this time is 4+. Also the paper is based on the data listed in Table 1. Which have been retrieved from smoothed aa index data. The smoothing could be done in many different ways each will produce different data sets. However, when considering the geomagnetic activities, we are usually interested to know how sever it will be and for how long it will last.
[Figure]

1) Therefore, I suggest the following. It should be stated clearly that these max values are for smoothed aa index and it should be given a special note. The paper title should also indicate this. R: Yes. Thank you. To clearly describe the data used, the title is changed to 'Estimating the maximum of 363-day-smoothing highest 3-hourly aa index in 3-day-interval by the preceding minimum of highest/lowest aa value for the 11-year solar cycle'.

2) The author could try to compare the expect strength of the 25 cycle with the previous cycles. So, we could understand is it will be more active or less active. R: Yes. We do. The strength of aaHmax for cycle 25 is estimated to be aaHmax(25)=85.5±6.9 (nT), about 32% stronger than that of cycle 24.

3) The author could try to predict a more reliable maximum of the aa index for the 25th cycle. To do so I could suggest to construct two data sets of the observed aa index minimum and maximum values for each 3 days or more. These two sets could be smoothed for 13 months. The correlation between these two data sets (for 3 days min and max values) are about 0.79. From these two data sets the author could peak the maximum and minimum aa index for each solar cycle and replace these values with those in Table 1. R: Yes. Thank you. In the revised manuscript, we used the 3-hourly aa index of ISGI since 1868. For each 3-day-interval, we find out the highest aa index (aaH) and the lowest aa index (aaL) from 24 values of the 3-hourly aa indices. Then, both aaH and aaL are smoothed by 363 days (121 points) to mimic the 13-month smoothing. The results are similar to those using 13-month smoothed monthly mean values, apart from that the maximum is estimated to be around 85 for the highest value. The results using 13-month smoothed monthly mean values are now retained and changed to Section 4, as suggested by Referee # 2.

4) Finally, the units of the indices (nT) should be written in text and on the Figures. R: Yes. We do.

[Figure]

2020.

---

## Author Comment (AC3) · 19 Jul 2020

Reply to Referee # 2 Overall modifications The manuscript has been thoroughly revised based on two referees. We discuss mainly the result using the 3-hourly aa index since 1868 in Sect 2. For each 3 days' interval, we find out the highest/lowest aa index (aaH/aaL) from 24 values of the 3-hourly aa indices. In order to reduce accidental events in the data, both aaH and aaL are smoothed by 363 days (121 points) to mimic the 13-month smoothing, as suggested by Referee 1. The maximum of aaH (aaHmax) is found to be well correlated to the preceding minimum of either aaH (aaHmin, r=0.85) or aaL (aaLmin, r=0.89) for the 11-year solar cycle. Based on these correlations, the strength of geomagnetic activity for cycle 25 is estimated to be aaHmax (25)= 85.5±6.9 (nT), similar to the average over the past cycles, but about 32% higher than that of

cycle 24. The rise time (THr) from aaHmin to aaHmax is found to be only weakly anti-correlated to the following aaHmax, r=-0.42. Such a weak correlation is no longer used to estimate THr as suggested by Referee 2. Similar result can also be obtained if using the 363-day-smoothing highest/lowest 3-hourly Ap index in 3-day-interval (ApH/ApL), shown in Sect. 3. The maximum of ApH (ApHmax) is well correlated to the preceding minimum of ApH (ApHmin, r=0.96) or ApL (ApLmin, r=0.79) for the 11-year solar cycle. The rise time (THa) from ApHmin to ApHmax is reversely correlated to the preceding minimum of ApL (ApLmin, r=-0.72). For the 13-month smoothed monthly mean aa (Ap) index, the result is moved down to Sect. 4, retained as a comparison, as suggested by Referee 2, but using only the aa index since 1868. The maximum aa(Ap) index, aamax (Apmax), of the solar cycle is also well correlated to the preceding minimum, aamin ( Apmin), with a correlation coefficient of r= 0.95(0.86). 'Predict' is changed to 'estimate' as suggested by Referee 2. ————————

The author identified maximum and minimums of smoothed aa and Ap indices through several solar cycles and calculate correlations between minimum and maximum values and between min/max values with respect to the preceding-following cycle. The relations that are found through the means of linear regression are then use to predict estimated aa/Ap minimum/maximum values for solar cycle 25.

Main comments 1. My main concern is with the selection of the dataset. It is not clear to me why the author choose to work with the 13-month smoothed aa index instead of the highest resolution available. Smoothing everything will naturally result in predictions that converge to the mean values and therefore fail to capture the spiky behavior of storm indices. This is particularly relevant in the case of Ap index. As shown in Table 1, the Ap smoothed monthly means corresponds to period of at most minor geomagnetic activity. Therefore all storm activity is lost. I suggest the author repeat the calculations using the highest available temporal resolution of the indices and compare them with the current results of the manuscript. R: Yes. We did. This suggestion is similar to that by Referee # 1: "To do so I could suggest to construct two

data sets of the observed aa index minimum and maximum values for each 3 days or more. These two sets could be smoothed for 13 months." In the revised manuscript, we used the 3-hourly aa index of ISGI since 1868 (highest resolution). For each 3-day-interval, we find out the highest aa index (aaH) and the lowest aa index (aaL) from 24 values of the 3-hourly aa indices. Then, both aaH and aaL are smoothed by 363 days (121 points) to mimic the 13-month smoothing, as suggested by Referee # 1. The results are similar to those using 13-month smoothed monthly mean values, apart from that the maximum is estimated to be around 85 for the highest value. The results using 13-month smoothed monthly mean values are now retained and changed to Section 4.

2 Page 2 L19-21 These results are hardly relevant. Simpler methods will estimate the duration of solar cycle phases with significantly better accuracy (For example, NOAA predicts a rising duration with an error of $\sim$8 months). Estimating the duration of half a cycle with an uncertainty of almost half the solar cycle results in a disconnection between the mathematical results and the known repetitiveness of the studies phenomena. There's a reason it is called the 11-year cycle. I suggest the author to revise the calculations and to interpret them in the context of what could be a reasonable assumption of the duration of the phases of SC25. R: As the anti-correlation coefficient between the rise time and the following maximum is very weak, we do no longer use it to estimate the rise time. The rise time of aa index is defined as the time duration from the minimum to the following maximum of aa/Ap index. The weak correlation between the rise time and the following maximum is related to the fact that the geomagnetic activity minimum (maximum) is not aligned to the solar (sunspot) activity minimum (maximum) in time, as shown in Fig.10 for the time difference of aamax to Rmax, $\Delta$Tmax(a), and that of aamin to Rmin, $\Delta$Tmin (b). In most cases, aamax (aamin) lags behind Rmax (Rmin). But in some other cases, aamax (aamin) precedes Rmax (Rmin). If the rise time is computed from the minimum of sunspot activity to aaHmax, the correlation is even weaker, r=-0.14.

3. The main results of the paper (shown in Figures 1-4) are heavily influenced by

the decision of using smoothed indices. While they may be correct in that particular context, the author should consider if the methodology utilized is the appropriate for this particular problem. Going back to point 1, if a different dataset is utilized, all figures need to be remade. On a note regarding presentation of the figures, adding colors to the different lines and making the figures of the appropriate size will significantly improve the readability. R: Yes. The dataset is replaced by the 3-hourly aa index of ISGI since 1868 as stated in 1. The rise time is defined as in 2. The lines of figures are shown in colors.

Specific comments 1) Title: I suggest replacing "minimum" with "solar minimum" to explicitly refer to solar cycle. Note that currently the title is misleading, as the prediction is regarding the smoothed data. Please adjust accordingly. R: Yes. We did. To clearly describe the data used, the title is changed to 'Estimating the maximum of 363-day-smoothing highest 3-hourly aa index in 3-day-interval by the preceding minimum of highest/lowest aa value for the 11-year solar cycle'. 2) Page 2 L10 - What is the meaning of a double plus-minus. Is it referring to different error sources? In that case please specify. L18 – Do you mean anti-correlated? L26 What do you mean by deviations? Please provide relevant references. R: The double plus-minus refers to different error sources, 'where $\pm 3.9$ and $\pm 2.1$ are derived from the uncertainty of aaHmax (25) and the standard deviation of the fitting of Ap to aa, respectively.' Yes. Thank you. It means anti-correlated. It changed to 'deviations of orbital motions of Satellites' (Yoshida and Yamagishi, 2010; Petrovay, 2020).

3) Page 4 L2-3 Predicted or estimated? A prediction is a statement about the future. A correlation between two variables at most indicates the ability to estimate one when the other is available, which appear to be the case. R: Yes. We changed 'estimated' to 'estimated'.

4) Page 7 L2-4 This extremely high correlation is clearly affected by the process of smoothing the data. Similar with other figures and equations, please correct based on major comments. R: Yes. We do. This figure is replaced by Fig.7(a) for the scat-

ter plot of the 363-day-smoothing 3-hourly Ap against aa indices since 1932 (dots). The correlation coefficient between them is r=0.93 (or 0.75 if using the non-smoothed series).
* * *

---

## Author Response (AR1)

**Reply to two referees**

Overall modifications

The manuscript has been thoroughly revised based on two referees.

To clearly describe the data used, the title is changed to 'Estimating the maximum of 363-day-smoothing highest 3-hourly aa index in 3-day-interval by the preceding minimum of highest/lowest aa value for the 11-year solar cycle'.

In the current version, we use the 3-hourly aa index since 1868 in Sect 2. For each 3 days' interval, we find out the highest/lowest aa index ($aa_H$/$aa_L$) from 24 values of the 3-hourly aa indices. In order to reduce accidental events in the data, both $aa_H$ and $aa_L$ are smoothed by 363 days (121 points) to mimic the 13-month smoothing, as suggested by Referee 1. The maximum of $aa_H$ ($aa_{Hmax}$) is found to be well correlated to the preceding minimum of either $aa_H$ ($aa_{Hmin}$, r=0.85) or $aa_L$ ($aa_{Lmin}$, r=0.89) for the 11-year solar cycle. Based on these correlations, the strength of geomagnetic activity for cycle 25 is estimated to be $aa_{Hmax}$ (25)= 84.5$\pm$6.9 (nT), similar to the average over the past cycles, but about 30% higher than that of cycle 24. The rise time ($T_{Hr}$) from $aa_{Hmin}$ to $aa_{Hmax}$ is found to be only weakly anti-correlated to the following $aa_{Hmax}$, r=-0.42. Such a weak

correlation is no longer used to estimate $T_{Hr}$ as suggested by Referee 2.

Similar result can also be obtained if using the 363-day-smoothing highest/lowest 3-hourly Ap index in 3-day-interval ($Ap_H$/$Ap_L$), shown in Sect. 3. The maximum of $Ap_H$ ($Ap_{Hmax}$) is well correlated to the preceding minimum of $Ap_H$ ($Ap_{Hmin}$, r=0.96) or $Ap_L$ ($Ap_{Lmin}$, r=0.79) for the 11-year solar cycle. The rise time ($T_{Ha}$) from $Ap_{Hmin}$ to $Ap_{Hmax}$ is weakly anti-correlated to the following maximum ($Ap_{Hmax}$, r=-0.33), but reversely correlated to the preceding minimum of $Ap_L$ ($Ap_{Lmin}$, r=-0.72).

For the 13-month smoothed monthly mean aa (Ap) index, the result is moved down to Sect. 4, retained as a comparison, as suggested by Referee 2, but using only the aa index since 1868. The maximum aa(Ap) index, $aa_{max}$ ($Ap_{max}$), of the solar cycle is also well correlated to the preceding minimum, $aa_{min}$ ($Ap_{min}$), with a correlation coefficient of r= 0.95(0.86).

'Predict' is changed to 'estimate' as suggested by Referee 2.

The sizes of figures are increased appropriately.

**Reply to Referee # 1**

The author uses smoothed monthly aa/Ap index to study the relation between the minimum and maximum aa/Ap values in order to predict the maximum value of the aa/Ap index. Usually long-term smoothing is used to study the solar cycle; to show the correlation between the solar cycle and the index variations. Due to the small number of high amplitude values, smoothing removes all the high amplitude maxima and move the data towards the minimum. The author states that "the maximum aa index for the ensuing cycle 25 is predicted to be aamax(25) = 26.9$\pm$2.6." This is very small value and it could be mistakenly understood that this solar cycle will be very quiet. The values listed in Table 1 under the aamax are much smaller than those observed in any disturbed day. These values can't represent the maximum aa index or the strength of the geomagnetic activities. As it could be seen from Fig. 1 the aa index has arrive to a peak value of about 67 nT in 19 March 2020 and the Kp value for this time is 4+. Also the paper is based on the data listed in Table 1. Which have been retrieved from smoothed aa index data. The smoothing could be done in many different ways each will produce different data sets. However, when considering the geomagnetic activities, we are usually interested to know how sever it will be and for how long it will last.

1) Therefore, I suggest the following. It should be stated clearly that these max values are for smoothed aa index and it should be given a special note. The paper title should also indicate this.

**R: Yes. Thank you.**

**The title is changed to 'Estimating the maximum of 363-day-smoothing highest 3-hourly aa index in 3-day-interval by the preceding minimum of highest/lowest aa value for the 11-year solar cycle'.**

2) The author could try to compare the expect strength of the 25 cycle with the previous cycles. So, we could understand is it will be more active or less active.

**R: Yes. We do.**

**The strength of $aa_{Hmax}$ for cycle 25 is estimated to be $aa_{Hmax}(25)=84.5\pm6.9$ (nT), about 30% stronger than that of cycle 24.**

3) The author could try to predict a more reliable maximum of the aa index for the 25th cycle. To do so I could suggest to construct two data sets of the observed aa index minimum and maximum values for each 3 days or more. These two sets could be smoothed for 13 months. The correlation between these two data sets (for 3 days min

and max values) are about 0.79. From these two data sets the author could peak the maximum and minimum aa index for each solar cycle and replace these values with those in Table 1.

**R: Yes. Thank you.**

**In the revised manuscript, we used the 3-hourly aa index of ISGI since 1868. For each 3-day-interval, we find out the highest aa index (aa$_H$) and the lowest aa index (aa$_L$) from 24 values of the 3-hourly aa indices. Then, both aa$_H$ and aa$_L$ are smoothed by 363 days (121 points) to mimic the 13-month smoothing. The results are similar to those using 13-month smoothed monthly mean values, apart from that the maximum is estimated to be around 84.5 for the highest value.**

**The results using 13-month smoothed monthly mean values are now retained and changed to Section 4, as suggested by Referee # 2.**

4) Finally, the units of the indices (nT) should be written in text and on the Figures.

**R: Yes. We do.**

**Reply to Referee # 2**

The author identified maximum and minimums of smoothed aa and Ap indices through several solar cycles and calculate correlations between minimum and maximum values and between min/max values with respect to the preceding-following cycle. The relations that are found through the means of linear regression are then use to predict estimated aa/Ap minimum/maximum values for solar cycle 25.

Main comments

1. My main concern is with the selection of the dataset. It is not clear to me why the author chooses to work with the 13-month smoothed aa index instead of the highest resolution available. Smoothing everything will naturally result in predictions that converge to the mean values and therefore fail to capture the spiky behavior of storm indices. This is particularly relevant in the case of Ap index. As shown in Table 1, the Ap smoothed monthly means corresponds to period of at most minor geomagnetic activity. Therefore, all storm activity is lost. I suggest the author repeat the calculations using the highest available temporal resolution of the indices and compare them with the current results of the manuscript.

**R: Yes. We did. This suggestion is similar to that by Referee # 1: "To do**

so I could suggest to construct two data sets of the observed aa index minimum and maximum values for each 3 days or more. These two sets could be smoothed for 13 months."

In the revised manuscript, we used the 3-hourly aa index of ISGI since 1868 (highest resolution). For each 3-day-interval, we find out the highest aa index ($aa_H$) and the lowest aa index ($aa_L$) from 24 values of the 3-hourly aa indices. Then, both $aa_H$ and $aa_L$ are smoothed by 363 days (121 points) to mimic the 13-month smoothing, as suggested by Referee # 1. The results are similar to those using 13-month smoothed monthly mean values, apart from that the maximum is estimated to be around 84.5 for the highest value.

The results using 13-month smoothed monthly mean values are now retained and changed to Section 4.

2 Page 2 L19-21

These results are hardly relevant. Simpler methods will estimate the duration of solar cycle phases with significantly better accuracy (For example, NOAA predicts a rising duration with an error of $\sim$8 months). Estimating the duration of half a cycle with an uncertainty of almost half the solar cycle results in a disconnection between the mathematical results and the known repetitiveness of the studies phenomena. There's a reason it is called the 11-year cycle. I suggest the

author to revise the calculations and to interpret them in the context of what could be a reasonable assumption of the duration of the phases of SC25.

**R: As the anti-correlation coefficient between the rise time and the following maximum is very weak (-0.42 for aa or -0.33 for Ap), we do no longer use it to estimate the rise time.**

**The rise time of aa index is defined as the time duration from the minimum to the following maximum of aa/Ap index. The weak correlation between the rise time and the following maximum is related to the fact that the geomagnetic activity minimum (maximum) is not aligned to the solar (sunspot) activity minimum (maximum) in time, as shown in Fig.10 for the time difference of $aa_{max}$ to $R_{max}$, $\triangle T_{max}$(a), and that of $aa_{min}$ to $R_{min}$, $\triangle T_{min}$ (b). In most cases, $aa_{max}$ ($aa_{min}$) lags behind $R_{max}$ ($R_{min}$). But in some other cases, $aa_{max}$ ($aa_{min}$) precedes $R_{max}$ ($R_{min}$). If the rise time is computed from the minimum of sunspot activity($R_{min}$) to $aa_{Hmax}$, the correlation is even weaker, r=-0.14.**

3. The main results of the paper (shown in Figures 1-4) are heavily influenced by the decision of using smoothed indices. While they may be correct in that particular context, the author should consider if the methodology utilized is the appropriate for this particular problem. Going back to point 1, if a different dataset is utilized, all figures need to

be remade. On a note regarding presentation of the figures, adding colors to the different lines and making the figures of the appropriate size will significantly improve the readability.

**R: Yes. The dataset is replaced by the 3-hourly aa index of ISGI since 1868 as stated in 1.**

**The rise time is defined as in 2.**

**The lines of figures are shown in colors and whose sizes are enlarged.**

Specific comments

1) Title: I suggest replacing "minimum" with "solar minimum" to explicitly refer to solar cycle. Note that currently the title is misleading, as the prediction is regarding the smoothed data. Please adjust accordingly.

**R: Yes. We did.**

**To clearly describe the data used, the title is changed to 'Estimating the maximum of 363-day-smoothing highest 3-hourly aa index in 3-day-interval by the preceding minimum of highest/lowest aa value for the 11-year solar cycle'.**

2) Page 2

L10 - What is the meaning of a double plus-minus. Is it referring to different error sources? In that case please specify. L18 – Do you mean anti-correlated? L26 What do you mean by deviations? Please provide

relevant references.

**R:** L10: **the double plus-minus refer to different error sources, 'where**

$\pm$**3.9 and** $\pm$**2.1 are derived from the uncertainty of aa$_{Hmax}$ (25) and**

**the standard deviation of the fitting of Ap to aa, respectively.'**

L18: **yes. Thank you. It means anti-correlated.**

L26: i**t changed to 'deviations of orbital motions of Satellites'**

**(Yoshida and Yamagishi, 2010; Petrovay, 2020).**

3) Page 4

L2-3 Predicted or estimated? A prediction is a statement about the future. A correlation between two variables at most indicates the ability to estimate one when the other is available, which appear to be the case.

**R: Yes. We changed 'estimated' to 'estimated'.**

4) Page 7

L2-4 This extremely high correlation is clearly affected by the process of smoothing the data. Similar with other figures and equations, please correct based on major comments.

**R: Yes. We do.**

[revised manuscript text omitted]

$$Ap_{\mathrm{Lmax}} = 1.8 \pm 1.1 + (0.10 \pm 0.04)Ap_{\mathrm{Hmin}}, \ \sigma = 0.7. \tag{7}$$

5  If $Ap_{\mathrm{Hmin}}$ is known, $Ap_{\mathrm{Lmax}}$ can be estimated from this equation.

**3.3 Relationship between the rise time and preceding minimum**

One may note in Table 4 that the anti-correlation between the rise time ($T_{\mathsf{Ha}}$) from $Ap_{\mathsf{Hmin}}$ to $Ap_{\mathsf{Hmax}}$ and the following maximum ($Ap_{\mathsf{Hmax}}$) is very weak, $r = -0.33$. While the anti-correlation between $T_{\mathsf{Ha}}$ and the preceding minimum ($Ap_{\mathsf{Lmin}}$) is strong, $r = -0.72$ (at the 95% confidence level). Figure 6(b) shows the scatter plot of $T_{\mathsf{Ha}}$ against $Ap_{\mathsf{Lmin}}$, fitted by the following linear equation,

$$T_{\mathsf{Ha}} = 9.6 \pm 1.4 - (2.3 \pm 0.9) Ap_{\mathsf{Lmin}}, \ \sigma = 1.6 \text{(years)}. \tag{8}$$

Similarly, if $Ap_{\mathsf{Lmin}}(25)$ is known, $T_{\mathsf{Ha}}$ for cycle 25 can be estimated from this equation.

**3.4 Relationship between $Ap$ and $aa$**

Now, we analyze the relationship between the $Ap$ and $aa$ indices, as shown in Fig. 7(a) for the scatter plot of the 363-day-smoothing 3-hourly $Ap$ against $aa$ indices since 1932 (dots). The solid line represents the linear fit of $Ap$ to $aa$ with the least-squares-fit regression equation given by

$$Ap = 0.12 \pm 0.01 + (0.5647 \pm 0.0005) aa, \ \sigma = 2.1. \tag{9}$$

The correlation coefficient between the fitted and observed values is $r = 0.93$ (or 0.75 if using the non-smoothed series) at a confidence level greater than 99%. It is obvious that $Ap$ is highly correlated with $aa$, as they are based on the same observations.

According to this equation, the maximum of 363-day smoothing highest $Ap$ value for cycle 25 can be estimated by substituting the estimated $aa_{\mathsf{Hmax}}(25) = 84.5 \pm 6.9$ (nT) in Sect. 2.2 into this equation, $Ap_{\mathsf{Hmax}}(25) = 47.8 \pm 3.9 \pm 2.1$ (nT), here $\pm 3.9$ is derived from the uncertainty ($\pm 6.9$) of $aa_{\mathsf{Hmax}}(25)$ and $\pm 2.1$ is the standard deviation of the fitting of $Ap$ to $aa$.

[Figure]

**Fig. 7.** (a) Scatter plot of the 363-day-smoothing 3-hourly $Ap$ against $aa$ indices since 1932 (dots) and the linear fit (solid). (b) The 13-month smoothed monthly mean time series of $aa$ (solid) since 1868 and $Ap$ (dotted) since 1932. The numbers in the figure indicate the solar cycles. The upper dashed and lower dash-dotted lines represent the maxima ($aa_{max}$) and minima ($aa_{min}$) of $aa$, respectively, for the 11-year solar cycle.

**Table 5.** Parameters of 13-month smoothed monthly mean $aa$ and $A_p$ for the solar cycle.

| $n$ | $aa_{min}$(nT) | $aa_{max}$(nT) | $T_r$(month) | $Ap_{min}$(nT) | $Ap_{max}$(nT) | $T_a$(month) |
|---|---|---|---|---|---|---|
| 11 | | 21.10 | | | | |
| 12 | 6.07 | 20.25 | 44 | | | |
| 13 | 10.77 | 23.66 | 23 | | | |
| 14 | 5.64 | 17.12 | 93 | | | |
| 15 | 8.26 | 22.60 | 63 | | | |
| 16 | 9.57 | 25.39 | 68 | | | |
| 17 | 12.06 | 24.66 | 64 | 7.29 | 16.82 | 112 |
| 18 | 15.26 | 28.56 | 79 | 9.78 | 22.45 | 82 |
| 19 | 15.34 | 31.42 | 62 | 10.55 | 18.64 | 56 |
| 20 | 12.56 | 29.07 | 109 | 7.37 | 18.81 | 111 |
| 21 | 15.33 | 32.51 | 31 | 10.37 | 20.08 | 29 |
| 22 | 16.18 | 32.20 | 51 | 9.62 | 20.24 | 57 |
| 23 | 14.69 | 32.90 | 72 | 8.11 | 19.65 | 72 |
| 24 | 7.85 | 19.33 | 67 | 3.84 | 11.72 | 71 |
| 25 | 12.78 | | | | | |
| Av. | 11.60 | 25.77 | 63.5 | 8.41 | 18.55 | 73.8 |

**4    Result for the 13-month smoothed monthly mean $aa/Ap$ index**

At last in this section, we simply analyze the previous result using the 13-month smoothed (with half weight at the two ends) monthly mean $aa$ index (solid) since 1868 and $Ap$ index (dotted) since 1932, as shown in Fig. 7(b). The upper dashed and lower dash-dotted lines represent the maximum ($aa_{max}$) and minimum ($aa_{min}$) of the $aa$ index, respectively, for the 11-year solar cycle. The parameters are listed in Table 5, in which, $T_r$ is the rise time from $aa_{min}$ to $aa_{max}$, $Ap_{max}$ and $Ap_{min}$ are the maximum and minimum of the $Ap$

index for the 11-year solar cycle, respectively, and $T_a$ is the rise time from $Ap_{min}$ to $Ap_{max}$.

[Figure]

**Fig. 8.** (a) Scatter plot of $aa_{max}$ against $aa_{min}$ (triangles) and the linear fit (solid). (b) Scatter plot of $Ap_{max}$ against $Ap_{min}$ (triangles) and the linear fit (solid).

**4.1 Relationship between $aa_{\text{max}}$ and $aa_{\text{min}}$**

Figure. 8(a) shows the scatter plot of $aa_{\text{max}}$ against $aa_{\text{min}}$ for cycles 11-24 (triangles). The solid line indicates the linear fit of $aa_{\text{max}}$ to $aa_{\text{min}}$ by the following equation,

$$aa_{\text{max}} = 10.4 \pm 1.7 + (1.36 \pm 0.14)aa_{\text{min}}, \ \sigma = 1.7. \tag{10}$$

The correlation coefficient between $aa_{\text{max}}$ and $aa_{\text{min}}$ is $r = 0.95$ (at a confidence level greater than 99%), **slightly higher than that,** $r = 0.85(0.89)$**, for the correlation between** $aa_{\text{Hmax}}$ **and** $aa_{\text{Hmin}}(aa_{\text{Lmin}})$ **in Fig. 2 using the 363-day-smoothing highest (lowest) 3-hourly** $aa$ **index in 3-day-interval.**

Substituting the value of $aa_{\text{min}}$ (12.78) for cycle $n = 25$ into this equation, one can estimate $aa_{\text{max}}(25) = 27.9 \pm 1.7$ (asterisk), about 44% higher than that (19.33) of cycle 24. **This estimate is similar to the case in Sect. 2.2 that the estimate (91.0) of** $aa_{\text{Hmax}}(25)$ **from** $aa_{\text{Lmin}}$ **in Eq. (2) is about 40% higher than that (64.81 nT) of cycle 24 using the minimum of 363-day-smoothing lowest 3-hourly** $aa$ **index in 3-day-interval.**

**4.2 Relationship between $Ap_{\text{max}}$ and $Ap_{\text{min}}$**

Figure 8(b) illustrates the scatter plot of $Ap_{\text{max}}$ against $Ap_{\text{min}}$ for cycles 17-24 (triangles). It is seen in the figure that $Ap_{\text{max}}$ is also well correlated to $Ap_{\text{min}}$, with a correlation coefficient of $r = 0.86$ (at a confidence level greater than 99%), **slightly lower (higher) than that,** $r = 0.96(0.79)$**, for the correlation between** $Ap_{\text{Hmax}}$ **and** $Ap_{\text{Hmin}}(Ap_{\text{Lmin}})$ **in Fig. 5 using the 363-day-smoothing highest (lowest) 3-hourly** $Ap$ **index in 3-day-interval.** The linear fitting equation of $Ap_{\text{max}}$ to $Ap_{\text{min}}$ (solid) is

$$Ap_{\text{max}} = 8.3 \pm 2.5 + (1.23 \pm 0.29)Ap_{\text{min}}, \ \sigma = 1.6. \tag{11}$$

If $Ap_{\text{min}}$ is known, $Ap_{\text{max}}$ can be estimated from this equation.

[Figure]

**Fig. 9.** (a) Scatter plot of $T_a$ against $aa_{max}$ (triangles) and the linear fit (solid). (b) Scatter plot of $T_r$ against $Ap_{max}$ (triangles) and the linear fit (solid).

**4.3 Relationship between the rise time and following maximum**

Figure 9(a) shows the scatter plot of the rise time ($T_r$) of $aa_{max}$ from $aa_{min}$ to $aa_{max}$ against the maximum ($aa_{max}$). It is seen in this figure that the data points are much

scattered, and so $T_r$ is nearly uncorrelated to the following $aa_{\mathsf{max}}$, $r = -0.09$. Similarly, the rise time ($T_{\mathsf{a}}$) of $Ap_{\mathsf{max}}$ from $Ap_{\mathsf{min}}$ to $Ap_{\mathsf{
[revised manuscript text omitted]

---

## Author Response (AR2)

**Reply to reviewer**

Thank you very much for valuable suggestions. The changes are indicated in **Boldface**.

(1) Figure 1 is inserted to show the non-smoothed series, for comparing with Fig.2 of the smoothed series and for the reason why we smooth it: **the average absolute differences of the adjacent values are 36.7 nT and 2.5 nT** (for $aa_H$ and $aa_L$), **respectively**. **It is hard to see the variation** in the non-smoothed series **with the solar cycle**.

(2) Equation (1) is inserted on Page 4 for **the running smoothing technique**. Equation (2) is inserted on Page 7 for **the 13-month (with half weight at the two ends) smoothed monthly mean sunspot number** ($R_I$).

(3) The analysis for the Ap index is removed, as suggested.

(4) The data are **updated to 10 October 2020**, $aa_{Hmin}$ is changed to **29.57** (from 31.05), and $aa_{Hmax}(25)$ is predicted to be **83.7$\pm$6.9(nT)**.

1. Title: I imagine there must be a middle ground between the original title and this very extensive title. I don't think it is necessary to fully describe the methodology in the title.

**R: Yes. The title is now changed to "Estimating the maximum of smoothed highest 3-hourly aa index in 3 days by the preceding minimum for the solar cycle".**

2. What do you mean mathematically by smoothing? It is a running average? A mean over X points? Also related:

**R: Equation (1) is inserted on Page 4 for the running smoothing technique.**

Page 4, L1: You don't explain why is relevant to mimic a 13-month smoothing.

Page 4, L15: What is an accidental event and why do you need to smooth over such a large period?

**R: Figure 1 is inserted on Page 5 for the non-smoothed series.**

**It is seen in the figure that both $aa_H$ and $aa_L$ vary dramatically: the average absolute differences of the adjacent values are 36.7 nT and 2.5 nT, respectively. It is hard to see the variation in $aa_H$ or $aa_L$ with the solar cycle. Especially, most (63%) values of $aa_L$ are the minimum (2 nT).**

**In order to analyze the long-term trend of aa with the solar cycle conventionally represented by the 13-month smoothed monthly mean sunspot number ($R_I$)…. The smooth-width is selected to be w=121 points (363 days) as it is close to that (1 year = 12 months) used in the smoothing of $R_I$.**

3. I'm still very confused about the naming of variables. It seems that the author determined the maximum and minimum of aa every three

days and smoothed the values over 121 points. Then they calculated the max/min of those values for every solar cycle and called that aa_Hmax, aa_Hmin and aa_Lmax, aa_Lmin, which are then the values used in the following sections to calculate correlations with sunspots. However, I can't find the place in the text where it is specified that aa_H,L_min,max are defined as the max over a solar cycle.

**R: Yes. The value of $aa_{Hmax}$ ($aa_{Lmax}$) is the maximum of $aa_H$ ($aa_L$) during the time period between two adjacent solar cycle minima ($R_{min}$) determined by the smoothed monthly mean $R_I$. The value of $aa_{Hmin}$($aa_{Lmin}$) is the minimum of $aa_H$ ($aa_L$) during the time period between two adjacent $aa_{Hmax}$ ($aa_{Lmax}$).**

4. The manuscript is very repetitive. Ap and aa exhibit a very high correlation (as the author correctly describes), but then all the calculations are done independently for both Ap and aa, reaching extremely similar conclusions (which is expected) adding an extra and maybe unnecessary layer of complexity to the reading of the manuscript.

**R: The analysis for the Ap index is removed for conciseness.**

**The 13-month smoothed monthly mean aa index is retained in analysis (and in Discussion), according to the previous suggestion.**

5. The writing and figures of the manuscript have significantly improved; however, I feel that the introduction is still confusing and need some

extra attention.

I'm suggesting some minor changes for the first section.

Abstract L1: "for an upcoming cycle" > "for an upcoming solar cycle"

Abstract L2: "service for planning future" > "service and for planning future"

Page 2, L21 (and other places through the manuscript): "geomagnetic activities" > "geomagnetic activity"

Page 2, L25: Solar cycle 24 already ended. Please update accordingly

Page 2, L26: "strengths" > "maximum strength" or maybe "maximum intensity"

Page 3, L3: "K indices" > "K index"?

R: **Thanks a lot.**

**"strengths" is changed to "maximum intensity".**

[revised manuscript text omitted]
_{\text{max}}$ to $R_{\text{m}}$ (a) and that ($\Delta T_{\text{min}}$) of $aa_{\text{min}}$ to $R_{\text{min}}$ (b)**. The intensity of geomagnetic activity can only be roughly evaluated from that of solar (sunspot) activity, as the linear correlation coefficient between the smoothed monthly mean $aa$ and $R_{\text{I}}$ is only 0.61 (Du, 2011c) or even lower (0.43) if using the non-smoothed series (Du, 2011b). In addition, the future solar activity is also unknown at the current time and so it can not

[Figure]

**Fig. 7.** The time difference between $aa_{\mathrm{max}}$ and $R_{\mathrm{m}}$ (a) and that between $aa_{\mathrm{min}}$ and $R_{\mathrm{min}}$ (b).

be directly used to estimate the future geomagnetic activity.

[revised manuscript text omitted]